# REASONING IN SPACE VIA GROUNDING IN THE WORLD

**Yiming Chen**[1,2,3]     **Zekun Qi**[4]     **Wenyao Zhang**[5,6]
**Xin Jin**[6]     **Li Zhang**[2,7*]     **Peidong Liu**[1*]

[1]Westlake University   [2]Shanghai Innovation Institute   [3]Zhejiang University
[4]Tsinghua University   [5]Shanghai Jiao Tong University   [6]Eastern Institute of Technology   [7]Fudan University

⌂ Project Page        Github Code        🤗 Huggingface

## ABSTRACT

In this paper, we claim that 3D visual grounding is one of the cornerstones of spatial reasoning and introduce the *Grounded-Spatial Reasoner (GS-Reasoner)* to explore the effective spatial representations that bridge the gap between them. Existing 3D LLMs suffer from the absence of a unified 3D representation capable of jointly capturing semantic and geometric information. This deficiency is manifested either in poor performance on grounding or in an excessive reliance on external modules, ultimately hindering the seamless integration of grounding and spatial reasoning. To address this, we propose a simple yet effective *dual-path pooling* mechanism that tightly aligns geometric features with both semantic and positional cues, constructing a unified image patch-based 3D representation that encapsulates all essential information without increasing the number of input tokens. Leveraging this holistic representation, GS-Reasoner is the first 3D LLM that achieves autoregressive grounding entirely without external modules while delivering performance comparable to state-of-the-art models, establishing a unified and self-contained framework for 3D spatial reasoning. To further bridge grounding and spatial reasoning, we introduce the *Grounded Chain-of-Thought (GCoT)* dataset. This dataset is meticulously curated to include both 3D bounding box annotations for objects referenced in reasoning questions and step-by-step reasoning paths that integrate grounding as a core component of the problem-solving process. Extensive experiments demonstrate that GS-Reasoner achieves impressive results on 3D visual grounding, which in turn significantly enhances its spatial reasoning capabilities, leading to state-of-the-art performance.

## 1 INTRODUCTION

Visual-spatial intelligence encompasses the capability to perceive, interpret, and reason about 3D spaces, including the spatial layouts, object sizes, positions and their potential interactions. This skill is fundamental to various domains, such as embodied intelligence and autonomous driving. Accurately linking 3D objects with textual descriptions, a task known as 3D visual grounding, is a prerequisite for effective spatial reasoning. This aligns with human cognitive processes, where identifying relevant objects is a fundamental step before reasoning about their spatial relationships. Despite recent advancements in 3D large language models (LLMs) (Cheng et al., 2024; Cai et al., 2024; Zhou et al., 2025; Zheng et al., 2025; Wang et al., 2025b; Hong et al., 2023a; Chen et al., 2024a; Huang et al., 2023b; 2024; Zhu et al., 2024b), 3D LLMs still rely on pretrained 3D detectors or external decoders for grounding. This reliance not only limits their ability to fully understand 3D scenes but also impedes the cohesive integration of grounding and spatial reasoning. Therefore, a critical question arises: **How can we enable 3D LLMs to perform natural and effective grounding in an autoregressive manner, thereby enhancing their spatial reasoning capabilities?**

We identify two primary challenges in grounding enhanced spatial reasoning. The first challenge arises from the inherent complexity of 3D data. Unlike 2D images, point cloud-based 3D scenes encode rich spatial relations and depth cues that are difficult to capture and align with the semantic space of LLMs, especially given the scarcity of large-scale 3D datasets. Moreover, representing

---

*Corresponding authors.

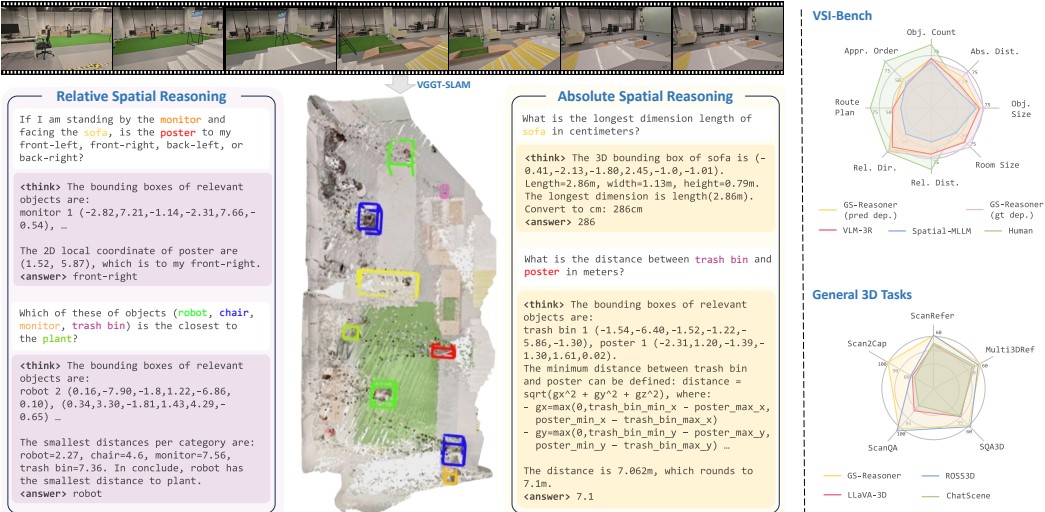

Figure 1: We propose *GS-Reasoner*, which integrates visual grounding as an intermediate chain-of-thought for spatial reasoning. All the bounding boxes shown above are autoregressively derived by GS-Reasoner in the reasoning process. Notably, the showcased video is captured in the wild without sensory 3D inputs, highlighting the strong generalization capability of our model.

such fine-grained structures often requires a substantially larger number of tokens, further increasing modeling cost. Previous works (Hong et al., 2023a; Chen et al., 2024a) compress point cloud features with Q-former, while others (Fu et al., 2025; Huang et al., 2025b) adopt voxel-based representations to better preserve structure. However, these methods typically trade geometric fidelity for token efficiency, and the extracted point cloud features contain only limited semantic information, making accurate grounding and reasoning difficult. More recent approaches (Zheng et al., 2025; Zhu et al., 2024b) encode 3D positional cues into video-based semantic features from vision foundation models, showing promising 3D reasoning benefits from visual LLM pretraining. Nevertheless, the geometric cues derived solely from 3D position encodings are weak, which constrains grounding performance. The second challenge lies in the lack of high-quality datasets that integrate grounding as an intermediate step for spatial reasoning. Existing 3D VQA datasets (Azuma et al., 2022; Ma et al., 2022) provide only short answers without grounding annotations or reasoning steps, making the combination of grounding and reasoning impossible. Additionally, these datasets fail to capture the contextual richness and structural complexity required for comprehensive spatial reasoning, further limiting progress toward robust 3D LLMs.

In this work, we propose a novel approach to address the identified challenges by introducing a comprehensive 3D scene representation and a GCoT dataset for spatial reasoning. Our 3D scene representation integrates semantic features from vision foundation models, geometric features encoded by a point cloud encoder, and 3D positional information. The key idea is to unify these heterogeneous signals within an image patch-based representation. Specifically, we pool the geometric features of point maps in a dual-path way to align them with the corresponding semantic feature and 3D position of the image patch, and subsequently fuse them into a unified hybrid representation. This hybrid representation preserves the strong generalization ability of LLMs gained from visual-semantic pretraining, while the incorporation of geometric information significantly strengthens its 3D scene comprehension. As a result, GS-Reasoner can accurately locate objects without relying on any external modules, which provides a natural intermediate step for spatial reasoning. To train models capable of handling both tasks, we construct the GCoT dataset. It includes precise 3D bbox annotations for objects mentioned in reasoning questions, along with step-by-step reasoning paths that embed grounding as a core component of problem solving. By structuring the tasks in this way, the dataset encourages models to first identify relevant objects before addressing complex spatial reasoning, yielding a more interpretable and cognitively aligned approach to learning spatial reasoning.

- We propose a semantic-geometric hybrid 3D scene representation that endows LLM with strong geometric priors, firstly enabling LLM to autoregressively perform 3D visual grounding with impressive results.

- We introduce the GCoT dataset, which bridges the gap between grounding and spatial reasoning, enabling GS-Reasoner to first ground objects and then reason about their spatial relationships in a manner aligned with human cognition.

- We demonstrate the effectiveness of GS-Reasoner through extensive experiments, showcasing its remarkable performance in both 3D visual grounding and spatial reasoning tasks.

## 2 RELATED WORK

**3D Large Language Models for 3D Understanding.** Recent advances in MLLMs have enabled 3D LLMs that integrate 3D information for tasks such as 3D VQA, visual grounding, and captioning. Early work 3D-LLM (Hong et al., 2023a) introduces a Q-Former to align point cloud features with LLMs, followed by studies (Chen et al., 2024a;b; Zhu et al., 2024a; Deng et al., 2025) constructing 3D representations with controllable token lengths. Voxel-based approaches (Fu et al., 2025; Huang et al., 2025b) balance token efficiency and geometric fidelity, while object-centric methods (Huang et al., 2024; 2025b; Yu et al., 2025) improve 3D scene understanding but lack global context. Recent works (Zheng et al., 2025; Zhu et al., 2024b; Wang et al., 2025b) propose encoding 3D positional information into visual features extracted by vision foundation models, achieving promising results on 3D tasks while maintaining the generalization ability of visual LLMs. Despite these advances, existing 3D LLMs still struggle to jointly capture semantic and geometric information from 3D scenes, limiting performance on 3D visual grounding or forcing reliance on external modules.

**Video-Language Models for Spatial Reasoning.** The goal of visual-based spatial intelligence is to equip video MLLMs with the ability to understand and reason about 3D spatial structures directly from video data. While Video-Language Models (VLMs) (Lin et al., 2023; Li et al., 2024; Bai et al., 2025; Liu et al., 2024; Chen et al., 2024c) perform well on video-language tasks, they still show limited results on recent spatial reasoning benchmarks (Yang et al., 2025). Spatial-MLLM (Wu et al., 2025a) and VLM-3R (Fan et al., 2025) enhance spatial reasoning by incorporating geometric features from recent developed visual geometry models (e.g., VGGT (Wang et al., 2025c)) and constructing large-scale spatial reasoning QA pairs for training. However, the constrained answer formats, such as single-choice selections or short numerical responses, potentially limit the ability of MLLMs to fully exploit the rich 3D information encoded in the geometric features of visual geometry models.

## 3 GS-REASONER FRAMEWORK

### 3.1 OVERVIEW

Given a sequence of $N$ RGB images $\{I_i \in \mathbb{R}^{3 \times H \times W}\}_{i=1}^{N}$ of a 3D scene and a spatial reasoning query $Q$, our goal is to build a model that can first identify all objects potentially relevant to $Q$ and then perform step-by-step spatial reasoning in an autoregressive manner to derive the final answer. Depth maps $\{D_i \in \mathbb{R}^{H \times W}\}_{i=1}^{N}$, camera intrinsics $K \in \mathbb{R}^{3 \times 3}$, and extrinsics $\{T_i \in \mathbb{R}^{4 \times 4}\}_{i=1}^{N}$ are assumed available or can be estimated using visual geometry methods (Maggio et al., 2025).

As illustrated in Fig. 2 (a) and (b), the proposed GS-Reasoner framework comprises three main components: a semantic encoder, a geometric encoder, and a video LLM. The semantic encoder extracts rich semantic features from the input RGB images using a pre-trained vision foundation model. Meanwhile, the depth maps are back-projected into point maps $\{P_i \in \mathbb{R}^{3 \times H \times W}\}_{i=1}^{N}$, which are subsequently transformed into geometric and 3D positional features. Specifically, the geometric encoder processes the aggregated sensor point cloud $\mathcal{P} = \cup_{i=1}^{N} P_i$, where $\mathcal{P} \in \mathbb{R}^{M \times 3}$ denotes $M$ 3D points, to capture structural information of the scene. Since the geometric features are permutation-invariant and thus lack explicit positional cues, we further position-encode the 3D coordinates of points. Finally, the semantic, geometric, and positional features are fused into a unified semantic-geometric hybrid 3D scene representation. This hybrid representation, together with the text query $Q$, is fed into the video LLM to perform autoregressive object grounding and spatial reasoning, ultimately producing the final answer.

We format the output of GS-Reasoner in a Chain-of-Thought (CoT) manner. All intermediate reasoning is enclosed within the "`<think>...</think>`" block: the model first analyzes the query, and then lists the 3D bounding boxes of all relevant objects in the following format "`OBJECT_NAME`

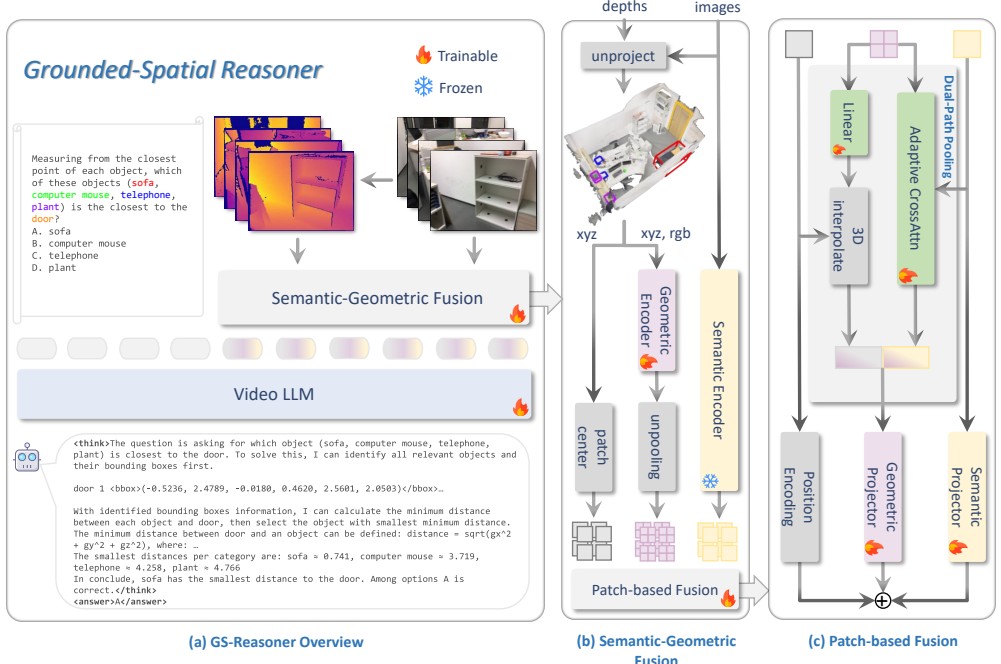

Figure 2: **Overview of GS-Reasoner framework**. Our method builds a semantic-geometric hybrid 3D scene representation, enabling 3D LLM to perform 3D visual grounding autoregressively, which allows grounding to be integrated as a chain-of-thought within the spatial reasoning process.

`OBJECT_COUNT <bbox>(x1, y1, z1, x2, y2, z2)</bbox>...`". If object bounding boxes are considered unhelpful for answering during question analysis, they are omitted. Each tuple "`(x1, y1, z1, x2, y2, z2)`" denotes the coordinates of two opposite corners of an axis-aligned 3D bounding box expressed in the world coordinate frame (units: meters). After grounding, the model carries out step-by-step spatial reasoning using all available information. Finally, it emits a concise final answer enclosed in "`<answer>...</answer>`". This autoregressive output format improves interpretability while remaining flexible, enabling GS-Reasoner to be applied to various 3D visual grounding and spatial-reasoning tasks without changing the architecture.

## 3.2 SEMANTIC-GEOMETRIC HYBRID 3D SCENE REPRESENTATION

In this section, we describe the construction of a comprehensive 3D scene representation that seamlessly integrates semantic and geometric information. Building on Video LLM, our goal is to enhance its spatial understanding capabilities by incorporating richer geometric cues, without increasing the input token count or compromising its language comprehension. Inspired by recent works (Zheng et al., 2025; Zhu et al., 2024b) that augment image patch features with 3D positional encoding, we design our 3D scene representation using image patches as the basic building block.

**Geometric Feature Extraction.** The first challenge arises when extracting per-patch geometric features from point maps. Instead of processing points independently within each patch—which often contain very few points and thus provide limited context for effective feature learning—we process the point cloud $\mathcal{P}$ as a whole. Specifically, we first partition the point maps $\{P_i \in \mathbb{R}^{3 \times H \times W}\}_{i=1}^{N}$ into patches of size $p \times p$, aligning with the image patch size used in the semantic encoder. To reduce computational cost, we uniformly sample $K$ points from each patch, resulting in subsampled point maps denoted as $\{P_i^{sub} \in \mathbb{R}^{3 \times K \times H' \times W'}\}_{i=1}^{N}$, where $H' = \frac{H}{p}$ and $W' = \frac{W}{p}$. The collection of all sampled points across patches forms the input point cloud $\mathcal{P}$, which is subsequently processed by a point cloud encoder to extract geometric features. We adopt the encoder-only Point Transformer v3 (PTv3) (Wu et al., 2024; 2025b) as our point cloud encoder owing to its efficiency and effectiveness. Given a point set $\mathcal{M} = (\mathcal{P}, \mathcal{F})$, where $\mathcal{F} \in \mathbb{R}^c$ denotes point attributes (e.g., position, color), PTv3 first serializes the input point cloud with space-filling curves and partitions the points into subsets $[\mathcal{M}_1, \mathcal{M}_2, ..., \mathcal{M}_{n'}]$ according to their serialization order. The serialized subsets are then processed

by a U-Net-like encoder-only architecture, where each layer employs serialized attention to capture both local and global context. Between layers, each subset $\mathcal{M}_i = (\mathcal{P}_i, \mathcal{F}_i)$ is pooled as follows,

$$\boldsymbol{f}_i' = \text{MaxPool}(\{\boldsymbol{f}_j \boldsymbol{U} \mid \boldsymbol{f}_j \in \mathcal{F}_i\}), \qquad \boldsymbol{p}_i' = \text{MeanPool}(\{\boldsymbol{p}_j \mid \boldsymbol{p}_j \in \mathcal{P}_i\}), \qquad (1)$$

where $(\boldsymbol{p}_i', \boldsymbol{f}_i')$ denotes the position and features of pooled point aggregated from subset $\mathcal{M}_i$, and $\boldsymbol{U} \in \mathbb{R}^{c \times c'}$ is a linear projection. Collecting pooled points from $n'$ subsets yields the point set $\mathcal{M}' = \{\boldsymbol{p}_i', \boldsymbol{f}_i'\}_{i=1}^{n'}$ for the next stage of encoding. Unpooling is performed by preserving mapping relationships through the pooling layers, which allows point features to be projected back to the original resolution and concatenated with features from the previous encoding stage as,

$$\boldsymbol{f}_i^{up} = \text{concat}(\boldsymbol{f}_i, \boldsymbol{f}_j^{up}), \qquad \text{if } (\boldsymbol{p}_i, \boldsymbol{f}_i) \in \mathcal{M}_j. \qquad (2)$$

By progressively unpooling across layers and mapping the features back to point maps, we obtain the final geometric feature maps $\{G_i \in \mathbb{R}^{C \times K \times H' \times W'}\}_{i=1}^N$, which are spatially aligned with the inputs.

**Dual-Path Pooling.** With extracted geometric feature maps $\{G_i \in \mathbb{R}^{C \times K \times H' \times W'}\}_{i=1}^N$ and point maps $\{P_i^{sub} \in \mathbb{R}^{3 \times K \times H' \times W'}\}_{i=1}^N$, a straightforward strategy for deriving per-patch representations is to apply max pooling within each patch on the geometric feature maps and mean pooling on the point maps, following the design in PTv3. However, we observed that this naive approach results in poor grounding performance, which can be attributed to two key issues: **(1) semantic-geometric misalignment**. While processing point cloud as a whole enhances the receptive field and enables more accurate geometric feature extraction compared to treating points within each patch independently, it also leads to misalignment between the geometric features and semantic features in each patch, as the 3D points in a patch can interact with almost all the points in point cloud, whereas the semantic features are constrained to the information visible in the current image. The geometric features pooled by max pooling emphasize the most salient features without considering the semantic context of the patch, exacerbating this misalignment. **(2) position-geometric misalignment**. Traditional point cloud encoders typically group points using KNN (Qi et al., 2017a;b; Zhao et al., 2021; Wu et al., 2022) or serialization (Wu et al., 2024; 2025b), ensuring that points within a group are spatially close in 3D space. This spatial proximity allows naive pooling strategies to effectively preserve geometric information within the group. In contrast, 3D points within an image patch do not necessarily satisfy this condition, particularly when a patch contains both foreground and background elements, which can lead to large spatial distances among points. Consequently, directly applying max pooling to the geometric features within the patch may introduce geometric inconsistencies, while mean pooling the 3D points can produce positions that are far from both foreground and background objects. These issues can negatively impact the accuracy of predicted 3D bounding boxes.

To address these challenges, we propose a simple yet effective dual-path feature fusion module that aligns semantic, geometric, and positional information at the patch level. To mitigate semantic-geometric misalignment, we construct semantic-aligned geometric features via a lightweight cross-attention network. Each patch's semantic feature serves as the query, while the $K$ geometric features within the patch serve as keys and values. The attention mechanism allows the network to selectively integrate the geometric features most relevant to the patch's semantic context. For the position-geometric misalignment, we directly sample the 3D point corresponding to each patch's center pixel for position encoding, and then interpolate the geometric features based on the position of the 3D point to obtain position-aligned geometric feature. This simple strategy ensures consistency between positional and geometric information: if the sampled point is on the foreground, the interpolated features mainly come from foreground points, and vice versa. Finally, the semantic-aligned and position-aligned geometric features are concatenated and projected to produce the final patch-level geometric features, which are then combined with the projected semantic feature and sampled 3D point positional encoding to obtain the final patch-level hybrid feature.

## 4 GCoT: GROUNDED CHAIN-OF-THOUGHT DATASET

Recent works (Wu et al., 2025a; Fan et al., 2025; Ouyang et al., 2025) attempt to improve the spatial reasoning ability of MLLMs by constructing large-scale QA pairs with 3D object annotations. However, the answers in these datasets are typically restricted to single choices or short numerical values. Such limited supervision narrows the learning space of MLLMs, thereby reducing their learning efficiency and resulting in less interpretable outcomes. In fact, spatial reasoning is largely

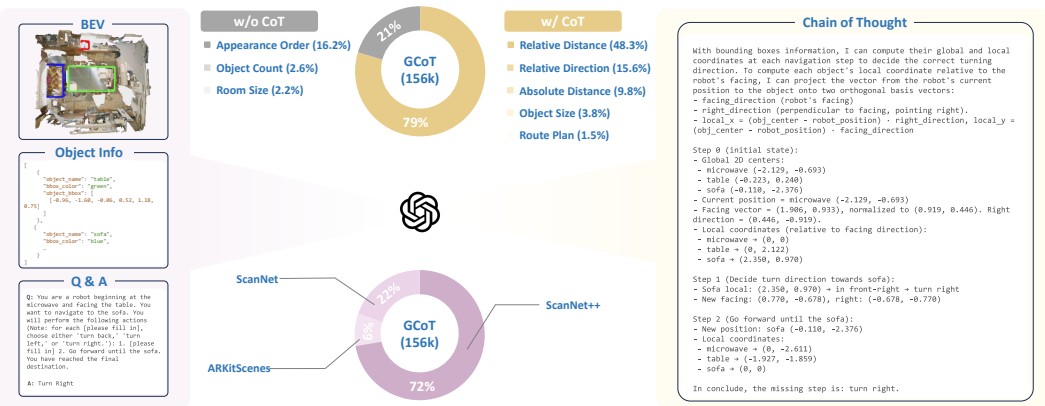

Figure 3: **Overview of Grounded Chain-of-Thought (GCoT) Dataset.** We first construct spatial QA pairs without CoT, and then prompt GPT-4o to generate CoT paths based on the bird's-eye view, object information, and QA pairs.

grounded in the locations and size relationships of relevant objects, indicating that identifying objects and reasoning about their geometry are fundamental steps, which is essentially a 3D visual grounding task. Introducing grounding as an intermediate step in spatial reasoning not only provides richer supervision but also improves interpretability, which motivates the construction of the GCoT dataset.

Fig. 3 presents an overview of the GCoT dataset. We first generate spatial reasoning QA pairs without CoT by following the dataset construction pipeline of (Yang et al., 2025; Fan et al., 2025), while preserving the object bounding box information used during generation. Leveraging the QA pairs, object bounding boxes, and bird's-eye views of the scenes, we then prompt GPT-4o (OpenAI et al., 2024) to produce coherent CoT reasoning paths that lead to the final answers. The resulting dataset consists of 156k QA pairs, among which 79% contain CoT annotations. We omit CoT construction for the Appearance Order, Object Counting, and Room Size Estimation tasks, as these tasks do not require complex spatial reasoning. Additional details are provided in the Appendix A.

## 5  EXPERIMENTS

We first describe the implementation details in Section 5.1 and report results on 3D visual grounding in Section 5.2. Since grounding forms the basis for spatial reasoning, we then evaluate our framework on spatial reasoning tasks in Section 5.3. Section 5.4 presents additional results on general 3D tasks, and Section 5.5 provides zero-shot evaluation and ablation studies to analyze the contributions of individual model components and the proposed GCoT dataset.

### 5.1  IMPLEMENTATION DETAILS

**Model Architecture.** GS-Reasoner is developed on top of LLaVA-Video 7B (Zhang et al., 2024), an open-source video LLM based on Qwen2-7B (Team, 2024). For semantic encoding, we adopt SigLIP (Zhai et al., 2023), a vision transformer pre-trained on large-scale image–text pairs through contrastive learning. For geometric encoding, we employ Sonata (Wu et al., 2025b), an efficient point cloud encoder built upon PTv3 (Wu et al., 2024) and pre-trained in a self-supervised manner on large-scale point cloud datasets. We adopt sinusoidal positional encoding (Vaswani et al., 2017) to encode the 3D positions of image patches.

**Training.** GS-Reasoner is trained end-to-end for next-token prediction. We first pretrain on subsets of 3D visual grounding datasets, including ScanRefer (Chen et al., 2020), Multi3DRef (Zhang et al., 2023), SR3D, and NR3D (Achlioptas et al., 2020), to warm up object grounding, and then finetune on our proposed GCoT dataset, the remaining grounding data, and other 3D tasks (ScanQA (Azuma et al., 2022), SQA3D (Ma et al., 2022), Scan2Cap (Chen et al., 2021)). Data augmentation is important for training GS-Reasoner and we provide more details in Appendix B.1.

**Inference.** Unless otherwise specified, we uniformly sample 32 images from each scene as the model input during inference. For the 3D visual grounding task, ground-truth depth maps and camera

Table 1: **Evaluation on 3D Visual Grounding.** GS-Reasoner achieves performance comparable to 3D LLMs using mesh proposals or external grounding, without any external components.

| Methods | ScanRefer | | Multi3DRef | | SR3D | NR3D |
|---|---|---|---|---|---|---|
| | Acc@25 | Acc@50 | F1@25 | F1@50 | Acc@25 | Acc@25 |
| *Expert Models* | | | | | | |
| 3D-VisTA (Zhu et al., 2023) | 51.0 | 46.2 | - | - | 56.5 | 47.7 |
| PQ3D (Zhu et al., 2024c) | 56.7 | 51.8 | - | - | 62.0 | 52.2 |
| UniVLG (Jain et al., 2025) | 63.5 | 56.4 | - | - | 73.0 | 56.3 |
| Locate 3D (McVay et al., 2025) | 61.1 | 50.9 | - | - | 68.2 | 56.1 |
| *3D LLMs + Proposals from Mesh PC* | | | | | | |
| Chat-Scene (Huang et al., 2024) | 55.5 | 50.2 | 57.1 | 52.4 | - | - |
| Inst3D-LMM (Yu et al., 2025) | 57.8 | 51.6 | 58.3 | 53.5 | - | - |
| Video-3D LLM (Zheng et al., 2025) | 58.1 | 51.7 | 58.0 | 52.7 | - | - |
| ROSS3D (Wang et al., 2025b) | 61.1 | 54.4 | 59.6 | 54.3 | - | - |
| SeeGround (Li et al., 2025b) | 44.1 | 39.4 | - | - | - | - |
| *3D LLMs + External Grounding Module* | | | | | | |
| Grounded 3D-LLM (Chen et al., 2024b) | 48.6 | 44.0 | 44.7 | 40.8 | - | - |
| ReGround3D (Zhu et al., 2024a) | 53.1 | 41.2 | - | - | - | - |
| LLaVA-3D (Zhu et al., 2024b) | 54.1 | 42.2 | 54.3 | 47.2 | - | - |
| *3D LLMs* | | | | | | |
| 3D-LLM (Hong et al., 2023b) | 30.3 | - | - | - | - | - |
| **GS-Reasoner** | 60.8 | 42.2 | 61.7 | 45.3 | 56.7 | 50.0 |

Table 2: **Evaluation on VSI-Bench.** GS-Reasoner achieves state-of-the-art performance on most tasks, with further gains using more accurate (ground-truth) depth.

| Methods | Rank. | Avg. | Numerical Question | | | | Multiple-Choice Question | | | |
|---|---|---|---|---|---|---|---|---|---|---|
| | | | Obj. Cnt. | Abs. Dist. | Obj. Size | Room Size | Rel. Dist. | Rel. Dir. | Route Plan | Appr. Order |
| *Baseline* | | | | | | | | | | |
| Chance Level (Random) | - | - | - | - | - | - | 25.0 | 36.1 | 28.3 | 25.0 |
| Chance Level (Frequency) | - | 34.0 | 62.1 | 32.0 | 29.9 | 33.1 | 25.1 | 47.9 | 28.4 | 25.2 |
| *VSI-Bench Perf. (†= Tiny Set)* | | | | | | | | | | |
| †Human Level | - | 79.2 | 94.3 | 47.0 | 60.4 | 45.9 | 94.7 | 95.8 | 95.8 | 100.0 |
| †Gemini-1.5 Flash | - | 45.7 | 50.8 | 33.6 | 56.5 | 45.2 | 48.0 | 39.8 | 32.7 | 59.2 |
| †Gemini-1.5 Pro | - | 48.8 | 49.6 | 28.8 | 58.6 | 49.4 | 46.0 | 48.1 | 42.0 | 68.0 |
| †Gemini-2.0 Flash | - | 45.4 | 52.4 | 30.6 | 66.7 | 31.8 | 56.0 | 46.3 | 24.5 | 55.1 |
| *Proprietary Models (API)* | | | | | | | | | | |
| GPT-4o | 3 | 34.0 | 46.2 | 5.3 | 43.8 | 38.2 | 37.0 | 41.3 | 31.5 | 28.5 |
| Gemini-1.5 Flash | 2 | 42.1 | 49.8 | 30.8 | 53.5 | 54.4 | 37.7 | 41.0 | 31.5 | 37.8 |
| Gemini-1.5 Pro | 1 | 45.4 | 56.2 | 30.9 | 64.1 | 43.6 | 51.3 | 46.3 | 36.0 | 34.6 |
| *Open-sourced VLMs* | | | | | | | | | | |
| InternVL2-40B | 3 | 36.0 | 34.9 | 26.9 | 46.5 | 31.8 | 42.1 | 32.2 | 34.0 | 39.6 |
| LongVILA-8B | 9 | 21.6 | 29.1 | 9.1 | 16.7 | 0.0 | 29.6 | 30.7 | 32.5 | 25.5 |
| VILA-1.5-40B | 7 | 31.2 | 22.4 | 24.8 | 48.7 | 22.7 | 40.5 | 25.7 | 31.5 | 32.9 |
| LongVA-7B | 8 | 29.2 | 38.0 | 16.6 | 38.9 | 22.2 | 33.1 | 43.3 | 25.4 | 15.7 |
| LLaVA-NeXT-Video-7B | 4 | 35.6 | 48.5 | 14.0 | 47.8 | 24.2 | 43.5 | 42.4 | 34.0 | 30.6 |
| LLaVA-NeXT-Video-72B | 1 | 40.9 | 48.9 | 22.8 | 57.4 | 35.3 | 42.4 | 36.7 | 35.0 | 48.6 |
| QWen2.5VL-7B | 5 | 33.0 | 40.9 | 14.8 | 43.4 | 10.7 | 38.6 | 38.5 | 33.0 | 29.8 |
| LLaVA-OneVision-7B | 6 | 32.4 | 47.7 | 20.2 | 47.4 | 12.3 | 42.5 | 35.2 | 29.4 | 24.4 |
| LLaVA-OneVision-72B | 2 | 40.2 | 43.5 | 23.9 | 57.6 | 37.5 | 42.5 | 39.9 | 32.5 | 44.6 |
| *Specialized Spatial Reasoning Models* | | | | | | | | | | |
| Spatial-MLLM-4B | 3 | 48.4 | 65.3 | 34.8 | 63.1 | 45.1 | 41.3 | 46.2 | 33.5 | 46.3 |
| VLM-3R-7B | 2 | 60.9 | 70.2 | 49.4 | 69.2 | 67.1 | 65.4 | 80.5 | 45.4 | 40.1 |
| **GS-Reasoner (pred dep.)** | 1 | 64.7 | 69.1 | 61.9 | 70.0 | 65.7 | 65.4 | 88.9 | 44.3 | 52.3 |
| GS-Reasoner (gt dep.) | - | 70.1 | 70.9 | 73.6 | 77.8 | 81.8 | 70.6 | 90.5 | 42.8 | 52.6 |

parameters are provided to ensure a fair evaluation. For the spatial reasoning task, depth maps and camera parameters are estimated using VGGT-SLAM (Maggio et al., 2025), followed by metric alignment with Moge2 (Wang et al., 2025e). More details are provided in Appendix B.2.

## 5.2 EVALUATION ON 3D VISUAL GROUNDING

We evaluate our model on four widely used 3D visual grounding benchmarks: ScanRefer, Multi3DRef, SR3D, and NR3D. For single-object grounding (ScanRefer, SR3D, NR3D), we report Acc@25 and Acc@50, where a prediction is correct if its Intersection over Union (IoU) with ground truth exceeds

Table 3: **Evaluation on General 3D Tasks.** GS-Reasoner outperforms state-of-the-art 3D LLMs on Scan2Cap and achieves comparable results on ScanQA and SQA3D.

| Methods | Scan2Cap | | | | ScanQA | | | | | SQA3D |
|---------|----------|------|--------|--------|--------|--------|-------|--------|------|-------|
| | B-4 ↑ | Rouge ↑ | CIDEr ↑ | Meteor ↑ | B-4 ↑ | Rouge ↑ | CIDEr ↑ | Meteor ↑ | EM ↑ | EM ↑ |
| 3D-LLM(flamingo) (Hong et al., 2023a) | - | - | - | - | 7.2 | 32.3 | 59.2 | 12.2 | 20.4 | - |
| 3D-LLM(BLIP2-flant5) (Hong et al., 2023a) | - | - | - | - | 12.0 | 35.7 | 69.4 | 14.5 | 20.5 | - |
| LL3DA (Chen et al., 2024a) | 36.8 | 55.1 | 65.2 | 26.0 | 13.5 | 37.3 | 76.8 | 15.9 | - | - |
| Chat-3Dv2 (Huang et al., 2023a) | - | - | - | - | 14.0 | - | 87.6 | - | - | 54.7 |
| LEO (Huang et al., 2023b) | 36.9 | 57.8 | 68.4 | 27.7 | 13.2 | 49.2 | 101.4 | 20.0 | 24.5 | 50.0 |
| Scene-LLM (Fu et al., 2025) | - | - | - | - | 12.0 | 40.0 | 80.0 | 16.6 | 27.2 | 54.2 |
| ChatScene (Huang et al., 2024) | 36.3 | 58.1 | 77.2 | 28.0 | 14.3 | 41.6 | 87.7 | 18.0 | 21.6 | 54.6 |
| LLaVA-3D (Zhu et al., 2024b) | 41.1 | 63.4 | 79.2 | 30.2 | 14.5 | 50.1 | 91.7 | 20.7 | 27.0 | 55.6 |
| Video-3D LLM (Zheng et al., 2025) | 42.4 | 62.3 | 83.8 | 28.9 | 16.2 | 49.0 | 102.1 | 19.8 | 30.1 | 58.6 |
| ROSS3D (Wang et al., 2025b) | 43.4 | 66.9 | 81.3 | 30.3 | 17.9 | 50.7 | 107.0 | 20.9 | 30.8 | 63.0 |
| **GS-Reasoner** | 47.6 | 69.2 | 101.0 | 32.1 | 16.2 | 49.2 | 102.6 | 19.8 | 29.9 | 59.9 |

0.25 or 0.5, respectively. For multi-object grounding (Multi3DRef), we use the F1 score computed at IoU thresholds of 0.25 and 0.5. For fair comparison, we group the baselines into four categories: (1) Expert Models, specifically designed for 3D grounding and trained with both bounding box and mask supervision; (2) 3D LLMs + Proposals from Mesh Point Cloud, which select from proposals generated by detectors such as Mask3D (Schult et al., 2022); (3) 3D LLMs + External Grounding Module, which pair a 3D LLM with an auxiliary grounding module; and (4) 3D LLMs, which directly predict bounding boxes autoregressively without relying on any external modules or proposals.

As shown in Tab. 1, our model achieves superior performance compared with 3D-LLM (Hong et al., 2023b). Among other 3D LLM-based methods, it achieves state-of-the-art F1@25 on Multi3DRef, even surpassing methods that rely on proposals or external grounding modules. Moreover, on ScanRefer Acc@50 and Multi3DRef F1@50, GS-Reasoner matches the performance of 3D LLMs with external grounding modules, despite using only noisy, incomplete sensor point clouds rather than high-quality mesh inputs. However, GS-Reasoner still lags behind 3D LLMs with proposals from mesh point clouds on these two metrics. We attribute this to two factors: (1) mesh point clouds are more complete and less noisy; and (2) conventional 3D detectors (e.g., Mask3D) are commonly trained with mask supervision, which is more conducive to precise object localization than bbox supervision, as also reported in (McVay et al., 2025; Jain et al., 2025). An interesting observation is that GS-Reasoner achieves comparable results to expert models on ScanRefer but falls behind on SR3D and NR3D, suggesting LLM-based methods are better at complex queries (as in ScanRefer), while expert models excel in precise localization for simpler descriptions (as in SR3D and NR3D).

## 5.3 EVALUATION ON SPATIAL REASONING

We evaluate GS-Reasoner's spatial reasoning capability on VSI-Bench (Yang et al., 2025), which comprises over 5,000 QA pairs from egocentric videos in ScanNet (Dai et al., 2017), ScanNet++ (Yeshwanth et al., 2023), and ARKitScenes (Baruch et al., 2021). VSI-Bench provides two answer formats, multiple choice (MCA) and numerical (NA), and covers eight tasks spanning spatial and temporal reasoning. We follow the official VSI-Bench evaluation protocol for metric computation. The results in Tab. 2 demonstrate that GS-Reasoner achieves impressive performance, particularly on the Relative Direction and Absolute Distance tasks, which require complex spatial reasoning and precise 3D object localization. It also attains state-of-the-art results on the Appearance Order task, indicating that our semantic-geometric hybrid features effectively preserve temporal information from the original video while providing additional spatial cues. Moreover, performance consistently improves with more accurate depth input, with the average score exceeding 70 and yielding nearly a 10-point gain over the previous state of the art.

## 5.4 EVALUATION ON GENERAL 3D TASKS

We further present results on three established 3D vision-language understanding tasks: Scan2Cap, ScanQA, and SQA3D, following the official protocols and reporting performance in terms of CIDEr, BLEU, METEOR, ROUGE, and exact-match (EM) accuracy. The results in Tab. 3 show that GS-Reasoner sets a new state of the art for 3D dense captioning, achieving the best results on Scan2Cap across all metrics and significantly surpassing the previous leading method, ROSS3D (Wang et al.,

Table 4: **Zero-shot 3D visual grounding.** We train the models exclusively on ScanNet and evaluate them on ScanNet++ and ARKitScenes for visual grounding, reporting all results in Acc@25. Note that GPT-4o is prompted to do 2D visual grounding and then back-project to 3D via depth map. Locate 3D is an expert model.

| Methods | ScanRefer | LX3D | |
|---|---|---|---|
| | ScanNet | ScanNet++ | ARKitScenes |
| GPT-4o VLM | 59.9 | 60.5 | 26.8 |
| Locate 3D | 61.1 | 56.7 | 46.2 |
| **GS-Reasoner** | 60.8 | 51.0 | 45.6 |

Table 5: **Ablation Study on Data Aug. and 3D Representation.** We train models to autoregressively predict 3D bounding box coordinates using ScanRefer and Multi3DRef, and report results on ScanRefer.

| Methods | Data Aug. | Pos. Enc. | Geo.Pool. | Acc@25 | Acc@50 |
|---|---|---|---|---|---|
| LLaVA-NeXT | ✗ | ✗ | ✗ | 0.0 | 0.0 |
| Video-3D LLM | ✗ | Avg | ✗ | 15.4 | 3.5 |
| | ✓ | Avg | ✗ | 53.2 | 29.8 |
| | ✓ | Avg | Max | $57.5_{+4.3}$ | $35.7_{+5.9}$ |
| | ✓ | Avg | Cross-Attn | $58.9_{+5.7}$ | $38.6_{+9.8}$ |
| | ✓ | Sample | Interpolate | $59.3_{+6.1}$ | $40.2_{+10.4}$ |
| **GS-Reasoner** | ✓ | Sample | Dual-Path | $\textbf{60.8}_{+7.6}$ | $\textbf{42.2}_{+12.4}$ |

Table 6: **Ablation Study on Grounded CoT Mechanism.** We report results only for tasks in the GCoT dataset that include CoT annotations, to highlight the effectiveness of grounded CoT.

| Methods | Avg. | Numerical Question | | Multiple-Choice Question | | |
|---|---|---|---|---|---|---|
| | | Abs. Dist. | Obj. Size | Rel. Dist. | Rel. Dir. | Route Plan |
| LLaVA-NeXT-Video ft (w/o CoT) | 52.3 | 45.1 | 64.3 | 58.9 | 60.7 | 32.5 |
| GS-Reasoner ft (w/o CoT) | $57.7_{+5.4}$ | $50.8_{+5.7}$ | $65.7_{+1.4}$ | $62.3_{+3.4}$ | $79.3_{+18.6}$ | $30.4_{-2.1}$ |
| **GS-Reasoner ft (Full)** | $\textbf{66.1}_{+13.8}$ | $\textbf{61.9}_{+16.8}$ | $\textbf{70.0}_{+5.7}$ | $\textbf{65.4}_{+6.5}$ | $\textbf{88.9}_{+28.2}$ | $\textbf{44.3}_{+11.8}$ |

2025b). We attribute these gains to explicitly predicting coordinates for 3D visual grounding, which forces the model to better capture geometric and positional cues, thereby improving dense captioning performance. However, GS-Reasoner does not achieve leading results on ScanQA and SQA3D. We believe the main reasons are the presence of many ambiguous questions in these datasets that do not clearly specify the target object, as well as a strong bias in answer distribution. These factors encourage the model to overfit to textual patterns instead of effectively exploiting 3D tokens. Recent studies (Huang et al., 2025a; Li et al., 2025a) have also shown that finetuning LLMs without 3D input can yield results comparable to the state of the art on ScanQA and SQA3D. Incorporating reconstruction constraints in the output (as done in ROSS3D (Wang et al., 2025b)) may help encourage the model to utilize 3D tokens, and we leave this for future research.

## 5.5 ANALYSIS AND ABLATION STUDIES

**Zero-shot Generalization.** We evaluate the zero-shot generalization of GS-Reasoner on unseen 3D scenes. The model is trained solely on ScanNet data (ScanRefer, SR3D, etc.), and tested on the ScanNet++ and ARKitScenes validation splits of the Locate3D dataset (McVay et al., 2025) without finetuning. As shown in Tab. 4, GS-Reasoner achieves performance comparable to SOTA expert models on ARKitScenes and demonstrates strong results in novel scene spatial reasoning (Fig. 1).

**Effectiveness of Data Augmentation and Semantic-Geometric Hybrid 3D Representation.** We conduct ablation studies to assess the effectiveness of our data augmentation strategies and the proposed semantic-geometric hybrid 3D representation, using the 3D visual grounding task as the evaluation benchmark. We believe this task directly reflects the model's ability to jointly leverage semantic and spatial information from the input 3D scene. The results in Tab. 5 show that the model fails to accurately predict 3D bbox coordinates when only image input is provided (LLaVA-Next). Incorporating average position encoding (as in Video-3D LLM) still results in poor performance due to overfitting. Data augmentation brings notable improvements, yet the model continues to struggle with precise object localization, as indicated by the low Acc@50. Finally, by introducing geometric features from the geometric encoder and employing Dual-Path Pooling to progressively fuse position-aligned and semantic-aligned geometric features, we achieve substantial gains in both Acc@25 and

Table 7: **Computation Cost Comparison.** All values are reported in milliseconds.

| Methods | 2D Vision Enc. | 3D Vision Enc. | Dual-Path | Total |
|---|---|---|---|---|
| LLaVA-NeXT-Video | 429 | - | - | 470 |
| **GS-Reasoner** | 430 | 204 | 41 | 737 |

Table 8: **Ablation Study on Input Frames.** We investigate the impact of different numbers of input frames and sampling strategies on 3D visual grounding performance.

| Frames Num | Sampling Strategy | ScanRefer | | Multi3DRef | |
|---|---|---|---|---|---|
| | | Acc@25 | Acc@50 | F1@25 | F1@50 |
| 32 | uniform | 60.8 | 42.2 | 61.7 | 45.3 |
| 48 | uniform | **61.7** | 44.5 | 62.2 | 46.9 |
| 64 | uniform | 61.2 | **45.1** | **62.5** | **48.0** |
| 32 | cdviews | 61.1 | 43.4 | 61.9 | 46.2 |

Acc@50. These results demonstrate that the proposed hybrid 3D representation strengthens the model's understanding of 3D scenes and enables more accurate visual grounding.

**Effectiveness of GCoT Dataset.** We also ablate the impact of incorporating grounding into the chain-of-thought process on spatial reasoning performance. Specifically, we remove the CoT part from the answers in the GCoT dataset and train the model to directly predict the answer from the 3D scene and question. We report results for five tasks that incorporate grounding within the CoT process in Tab. 6. The results show that integrating grounding as part of the CoT process substantially improves performance across all tasks, particularly in object absolute distance, object relative direction, and route planning. This highlights the importance of not only providing the model with grounding capabilities but also guiding it to leverage grounding effectively to support spatial reasoning, demonstrating the necessity of the proposed GCoT dataset.

**Computation Costs Evaluation.** GS-Reasoner's computational overhead compared to the backbone LLaVA-Next-Video primarily arises from the construction of visual tokens. Specifically, GS-Reasoner requires encoding point clouds using Point Transformer and integrating them into image patch features. Therefore, we conduct a comparison of computational overhead during the visual token construction phase, using a single 4090 GPU with 32 input images. As shown in Tab. 7, GS-Reasoner introduces some additional computational overhead. However, since all visual tokens only need to be constructed once per response, this overhead is acceptable relative to the overall inference time. Furthermore, during subsequent reasoning, the semantic-geometric hybrid 3D representation constructed by GS-Reasoner does not increase the number of visual tokens, so the inference time remains comparable to that of LLaVA-Next-Video.

**Ablation Study on Input Frames.** We conduct an ablation study to investigate the impact of different numbers of input frames and sampling strategies on 3D visual grounding performance. As shown in Tab. 8, model performance consistently improves with an increasing number of input images, and the improvements are more pronounced in Acc@50 (F1@50) compared to Acc@25 (F1@25), indicating that providing denser point clouds can indeed help enhance localization accuracy. Moreover, employing context-aware sampling strategies such as cdViews (Wang et al., 2025a) also yields certain improvements.

## 6 CONCLUSION

In this work, we present GS-Reasoner, a novel framework that integrates grounding into spatial reasoning as a chain-of-thought process. Built upon a hybrid semantic–geometric 3D scene representation, GS-Reasoner performs grounding without requiring any external modules, making it a natural intermediate step for spatial reasoning. The GCoT dataset further strengthens the model's ability to handle both tasks effectively.

## ACKNOWLEDGMENTS

This work was supported in part by New Generation Artificial Intelligence-National Science and Technology Major Project (2025ZD0123004), Ningbo grant (2025Z038) and National Natural Science Foundation of China (Grant No. 62376060).

## ETHICS STATEMENT

Our work introduces a new dataset generated entirely using LLMs. As the dataset is synthetically generated, it does not contain any personally identifiable information or sensitive human data. Nevertheless, synthetic data may inherit biases present in the underlying LLM, and could potentially be misused for harmful or misleading purposes. To mitigate these risks, the dataset is intended solely for academic research, and will be released with clear guidelines on responsible usage. Users are encouraged to consider ethical implications when employing the dataset for downstream tasks.

## REPRODUCIBILITY STATEMENT

To facilitate reproducibility, we will release the full dataset, preprocessing scripts, and detailed documentation upon acceptance. All experimental code, pretrained models, and evaluation protocols will also be made publicly available. The datasets used in our experiments are either publicly accessible or will be released as part of this work. We provide complete hyperparameter settings, training schedules, and random seeds in the paper and supplementary materials.

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

# A    ADDITIONAL DATASET DETAILS

Grounding Chain-of-Thought Dataset plays a crucial role in our training, guiding the model to learn how to incorporate 3D visual grounding as an intermediate step in spatial reasoning. Here, we provide additional details on the dataset construction.

## A.1    INSTRUCTION QA GENERATION

We first construct spatial reasoning QA pairs without chain-of-thought annotations, following the dataset generation pipeline of (Yang et al., 2025; Fan et al., 2025). All data are sourced from existing large-scale 3D datasets, including ScanNet (Dai et al., 2017), ScanNet++ (Yeshwanth et al., 2023), and ARKitScenes (Baruch et al., 2021). To ensure consistency across datasets, we perform the following preprocessing steps for each scene:

- **Point Cloud.** We directly use the raw point cloud provided by each dataset. Since ScanNet++ and ARKitScenes do not guarantee alignment of the global coordinate system with the physical room structure (e.g., the XY plane may not align with walls or floors), we further apply axis alignment. Specifically, we estimate the gravity direction and compute the principal components of the point cloud to align the axes, yielding a transformation matrix for each scene.
- **Sampled Frame Data.** we uniformly sample 50 RGB frames, which serve as the basis for constructing frame metadata. These frames provide a consistent visual context for generating spatial reasoning questions and ensure coverage of diverse viewpoints within the scene.

Based on the preprocessed data, we further construct detailed metadata for each scene, consisting of the following components:

- **Scene Metadata.** This metadata is used for all spatial reasoning questions construction. We extract the axis-aligned bounding boxes (AABBs) of all object instances, either directly from mask annotations or by converting from oriented bounding box (OBB) annotations. In addition to bounding boxes, this metadata also includes global scene statistics such as the number of objects and room dimensions, which are later used to formulate numerical reasoning questions.
- **Frame Metadata.** This metadata is used specifically for appearance-based temporal reasoning questions. For each object, we determine its appearance time by recording the first frame in which its 2D mask area exceeds a given threshold. Consequently, the frame metadata of each scene contains the appearance time of all objects, enabling the construction of reasoning questions grounded in temporal visual evidence.

These two types of metadata provide the necessary information to generate a diverse set of spatial and temporal reasoning questions. Following the predefined question templates in (Yang et al., 2025), we iterate over the scene metadata to construct a large pool of candidate questions and their corresponding answers. The detailed procedures are as follows:

**Spatial Reasoning QA.**

- **Object Count.** For each object category with at least two instances in the scene, we generate counting questions by directly querying the number of instances.
- **Absolute Distance.** We randomly select pairs of objects that appear only once in the scene and compute their Euclidean distance, which serves as the basis for absolute distance queries.
- **Object Size.** For objects with a single instance, we compute the object size using the diagonal length of their AABB, and use this value to construct size-related questions.
- **Room Size.** We estimate the overall room size of each scene using the alpha-shape algorithm applied to the scene point cloud, allowing us to ask questions about scene-level spatial dimensions.
- **Relative Distance.** We randomly select a set of $N$ objects ($3 \leq N \leq 5$), compute all pairwise distances, and identify the closest pair of objects. This enables the construction of questions that require comparative spatial reasoning.

- **Relative Direction.** We randomly select three objects with unique instances and compute their relative directions based on the centers of their AABBs. The resulting orientation relations form the basis of direction-based reasoning questions.

**Temporal Reasoning QA.** For temporal reasoning (i.e., appearance order), we randomly select four objects from each scene and determine their order of appearance using the frame metadata.

**Route Planning QA.** For route planning questions, we follow the procedure in VLM-3R (Fan et al., 2025) and employ the Habitat simulator to generate diverse navigation trajectories between two predefined points in each scene. The turning direction at each step is determined by computing the angle between consecutive anchor points along the trajectory. To identify relevant objects, we calculate their proximity to the trajectory by measuring the distance between anchor points and the 3D bounding boxes of scene objects provided in the scene metadata. Finally, we construct the QA pairs using predefined templates consistent with those in VLM-3R (Fan et al., 2025), where each question is grounded in the trajectory's turning direction and nearby objects.

## A.2    CHAIN-OF-THOUGHT GENERATION

Building upon the generated QA pairs and their associated 3D object bounding boxes, we further create CoT annotations to guide the model in exploiting 3D visual grounding for spatial reasoning. Specifically, we employ GPT-4o (OpenAI et al., 2024) to generate CoT reasoning paths for each QA pair. To mitigate potential hallucinations from the LLM, we construct a bird's-eye-view map of each scene using the preprocessed point cloud. On this map, we explicitly annotate the 3D bounding boxes of relevant objects, with different colors denoting distinct object categories, consistent with the color keys used in the input bounding box information. The BEV map, together with the question, answer, and relevant object annotations, is then provided to GPT-4o to generate coherent and detailed CoT paths. For efficiency, we exclude tasks that do not require complex spatial reasoning or grounding (e.g., Room Size, Object Count, Appearance Order) from CoT generation, as these can be accurately answered without the need for step-by-step grounding-based reasoning.

## A.3    DATA FORMAT

To train a model that can determine when to ground relevant objects and perform step-by-step reasoning autonomously, we prepend a *question analysis* part to each answer. This part explicitly guides the model to decide whether grounding is required before initiating the reasoning process. An example of the final data format for QA pairs with CoT is shown below:

---
**QA w/ CoT Example**

Question: [preprompt] If I am standing by the radiator and facing the table, is the telephone to the left or the right of the table? Options: A. left B. right

Answer: <think>The question is asking for the relative direction of the telephone with respect to my position (at the radiator) while facing the table. To solve this, I can identify all relevant objects and their bounding boxes first.

radiator 1 <bbox>(-1.9165, -1.0266, 0.0798, -1.6415, 0.9513, 0.6104)</bbox>, table 1 <bbox>(-0.9128, -1.7861, 0.0150, -0.5043, -1.0341, 0.5050)</bbox>, telephone 1 <bbox>(-1.4182, 0.9637, 0.9736, -1.1277, 1.2378, 1.0825)</bbox>

To determine the telephone's position relative to the table, I analyze spatial relationships through three steps:
Step 1: Observer Position and Orientation
Standing at the radiator's centroid:
- x = (-1.9165 + -1.6415)/2 = -1.779
- y = (-1.0266 + 0.9513)/2 = -0.038
Facing the table, whose centroid: ..
The telephone lies to the left of the table. Option A is correct. </think>
<answer>A</answer>

---

And example of the final data format for QA pairs without CoT is shown below:

---

**QA w/o CoT Example**

Question: [preprompt] What is the size of this room (in square meters)? If multiple rooms are shown, estimate the size of the combined space.

Answer: <think>The question is asking for the room size in square meters. To solve this, I can rely on the overall spatial information present in the video without identifying any object bounding boxes.
From the video, the room size is about 11.0 m$^2$.</think>
<answer>11.0</answer>

---

By structuring the data in this way, the model learns to autonomously decide when to ground relevant objects and perform step-by-step reasoning, without the need for additional prompting.

## B ADDITIONAL IMPLEMENTATION DETAILS

### B.1 TRAINING DETAILS

GS-Reasoner is trained end-to-end with cross-entropy loss for next-token prediction. We first pretrain on subsets of 3D visual grounding datasets, including ScanRefer (Chen et al., 2020), Multi3DRef (Zhang et al., 2023), SR3D, and NR3D (Achlioptas et al., 2020), to warm up object grounding, and then finetune on our proposed GCoT dataset, the remaining grounding data, and other 3D tasks (ScanQA (Azuma et al., 2022), SQA3D (Ma et al., 2022), Scan2Cap (Chen et al., 2021)). All parameters are learnable except those of the vision encoder. The LLM learning rate is fixed at $1e^{-5}$, while other modules use $1e^{-4}$ during pretraining and $1e^{-5}$ during finetuning. We use a batch size of 16 for pretraining and 256 for finetuning, set $K = 64$ in all experiments, and uniformly sample $N \in [16, 48]$ images per scene during training. Data augmentation is crucial for training GS-Reasoner, as the autoregressive objective tends to overfit object grounding under limited 3D data. We avoid conventional point cloud augmentations (e.g., jittering, elastic distortion) already covered in Sonata's pretraining, and instead focus on decoupling geometric and positional cues. Specifically, we apply Z-axis rotations of $[90°, 180°, 270°]$, random scaling within $[0.75, 1.25]$, and translations within $[-1, 1]$ meters, which alter bounding box positions and scales, forcing the model to exploit both cues for accurate predictions.

### B.2 INFERENCE DETAILS

We develop a pipeline to recover metric depth and camera parameters from multi-view images, enabling spatial reasoning without any input beyond images. Specifically, we first use VGGT-SLAM (Maggio et al., 2025) to reconstruct dense depth maps and relative camera intrinsics and extrinsics from the multi-view images. We then apply MoGe-2 (Wang et al., 2025e) to estimate absolute-scale depth maps and per-image camera intrinsics independently. Since the intrinsics from these two methods may not be aligned, we avoid direct scale estimation in the depth dimension. Instead, we project all points into the camera coordinate system and compute a global scale factor $s$ such that the scaled VGGT-SLAM point maps align with the corresponding MoGe-2 point maps across all views. Formally, $s$ is obtained by solving the following optimization problem:

$$s^* = \arg\min_{s>0} \sum_{i=1}^{N} \sum_{j=1}^{M_i} \| s \cdot p_{i,j}^{\text{VGGT-SLAM}} - p_{i,j}^{\text{MoGe-2}} \|^2, \tag{3}$$

where $p_{i,j}^{\text{VGGT-SLAM}}$ and $p_{i,j}^{\text{MoGe-2}}$ denote the $j$-th point in the $i$-th view from VGGT-SLAM and MoGe-2, respectively, $M_i$ is the number of valid points in view $i$, and $N$ is the total number of views. Furthermore, we compute a per-scene axis-alignment transformation matrix based on the estimated camera poses and reconstructed point clouds.

Table 9: **Ablation study on the impact of predicted depth maps and poses on spatial reasoning.**

| Recon. Methods | Align Methods | Avg. | Numerical Question | | | | Multiple-Choice Question | | | |
|---|---|---|---|---|---|---|---|---|---|---|
| | | | Obj. Cnt. | Abs. Dist. | Obj. Size | Room Size | Rel. Dist. | Rel. Dir. | Route Plan | Appr. Order |
| - | - | 70.1 | 70.9 | 73.6 | 77.8 | 81.8 | 70.6 | 90.5 | 42.8 | 52.6 |
| VGGT-SLAM | MoGe-2 | 64.7 | 69.1 | 61.9 | 70.0 | 65.7 | 65.4 | 88.9 | 44.3 | 52.3 |
| VGGT-SLAM | GT-Depth | 67.8 | 69.4 | 68.9 | 76.5 | 78.8 | 65.2 | 87.6 | 43.4 | 52.9 |
| VGGT | MoGe-2 | 59.6 | 66.8 | 55.6 | 65.5 | 63.1 | 58.3 | 77.3 | 34.0 | 51.6 |
| CUT3R | - | 56.8 | 66.8 | 55.3 | 60.2 | 50.7 | 59.8 | 76.6 | 34.0 | 50.7 |
| TTT3R | - | 58.0 | 67.5 | 57.6 | 60.2 | 46.5 | 62.8 | 82.9 | 35.1 | 51.5 |

## C  ADDITIONAL RELATED WORK

**Point Cloud Representation Learning.** Point cloud representation learning has been extensively studied for 3D understanding. Early works like PointNet (Qi et al., 2017a;b) use MLPs and symmetric functions to extract global features from point clouds. More recent methods, such as Point Transformer (Zhao et al., 2021; Wu et al., 2022; Qi et al., 2023b; Wu et al., 2024; 2025b), leverage attention mechanisms to capture local geometric structures and point relationships. ACT (Dong et al., 2022) pioneers cross-modal geometry understanding through 2D or language foundation models such as CLIP (Radford et al., 2021) or MAE (He et al., 2022). RECON (Qi et al., 2023a; 2024) further proposes a learning paradigm that unifies generative and contrastive learning. Despite architectural differences, these methods share a common pipeline: points are grouped based on spatial distribution, features are extracted per group, and then aggregated into a global representation. The resulting sparse features can be upsampled to the original resolution for tasks such as semantic segmentation.

## D  ADDITIONAL EXPERIMENTAL ANALYSIS AND RESULTS

### D.1  ANALYSIS ON GENERAL 3D TASKS

As shown in Tab. 3, GS-Reasoner does not achieve leading results on ScanQA and SQA3D. We believe the main reasons are the presence of many ambiguous questions in these datasets that do not clearly specify the target object, as well as a strong bias in answer distribution. These factors encourage the model to overfit to textual patterns instead of effectively exploiting 3D tokens. Recent studies (Huang et al., 2025a; Li et al., 2025a) have also shown that finetuning LLMs without 3D input can yield results comparable to the state of the art on ScanQA and SQA3D. Incorporating reconstruction constraints in the output (as done in ROSS3D (Wang et al., 2025b)) may help encourage the model to utilize 3D tokens, and we leave this for future research.

### D.2  ABLATION STUDY ON PREDICTED DEPTH MAPS AND POSES

We further evaluate the impact of different depth and pose estimation methods on the final spatial reasoning performance. Specifically, we consider the following geometry reconstruction methods: VGGT-SLAM (Maggio et al., 2025), VGGT (Wang et al., 2025c), CUT3R (Wang et al., 2025d), and TTT3R (Chen et al., 2025). For relative geometry reconstruction methods (VGGT-SLAM, VGGT), we use MoGe-2 (Wang et al., 2025e) for absolute scale recovery. We also experiment with using ground-truth depth for absolute scale recovery. The results are shown in Tab. 9, where the first row corresponds to using ground-truth depth and poses. From the experiments, we summarize two key factors that influence performance:

- **Accuracy of metric scale.** As observed, the most significant performance drop compared to using ground truth depth occurs in tasks related to absolute size estimation, such as room size and object absolute distance. During training, GS-Reasoner utilizes ground truth depth, which leads the model to prioritize 3D features in the input for these tasks. Consequently, inaccuracies in metric scale have a substantial impact on the results of such tasks. Notably, when using ground truth depth instead of MoGe2-predicted depth for metric alignment, there is a significant performance improvement.

- **Accuracy of pose.** The accuracy of pose affects the relative positional distribution of object point clouds within the scene, thereby influencing the model's understanding of spatial relationships between objects. In the point clouds estimated by CUT3R, we observed significant pose errors,

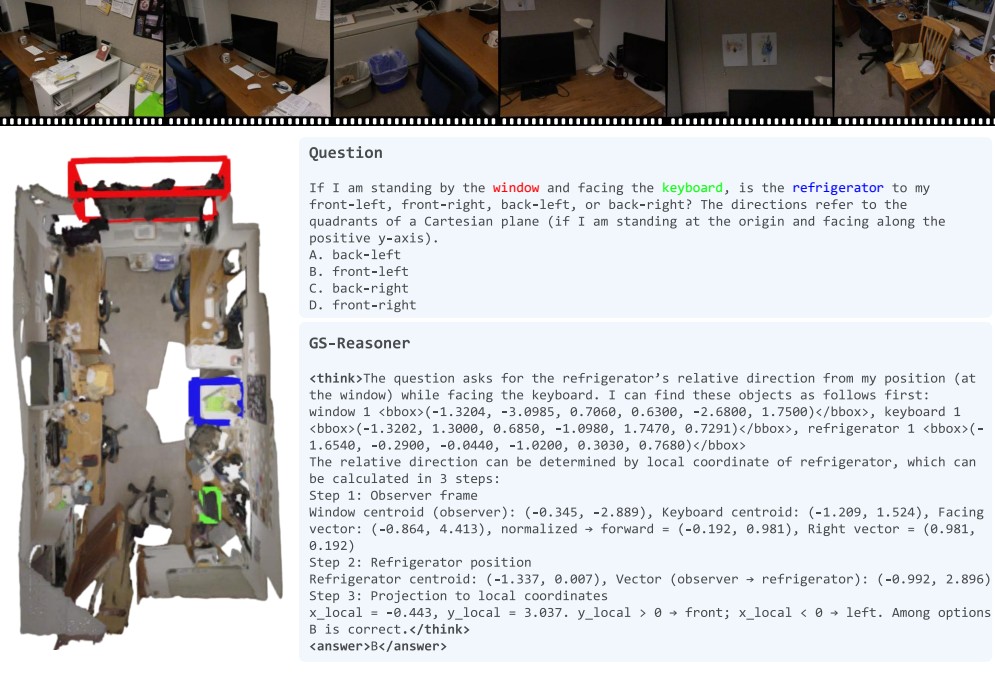

Figure 4: **Qualitative results on VSI-Bench.**

often resulting in completely incorrect scene layouts. This leads to a noticeable decline in GS-Reasoner's performance on tasks such as object relative direction. TTT3R improves upon CUT3R by reducing pose error accumulation during long-sequence inputs, resulting in performance gains.

Beyond these two factors, other aspects have a relatively minor impact on spatial reasoning. For instance, the quality of depth details (e.g., sharpness) does not significantly affect GS-Reasoner.

### D.3 MORE QUALITATIVE RESULTS

We present qualitative results of GS-Reasoner on VSI-Bench (Yang et al., 2025) as follows:

## E FUTURE WORK

Spatial reasoning is a key aspect of robotics and embodied reasoning, especially for the vision-language-action (VLA) models (Kim et al., 2024; Qi et al., 2025; Zhang et al., 2025). Leveraging the strong spatial reasoning ability of GS-Reasoner in robotic tasks can substantially enhance the generalization and robustness of embodied reasoning. Future directions include jointly fine-tuning with GCoT data and action data, and employing GS-Reasoner as an embodied brain for planning and task decomposition.

## F THE USE OF LARGE LANGUAGE MODELS (LLMS)

In this work, we leverage LLMs to facilitate the construction of our *Grounded Chain-of-Thought (GCOT)* dataset. Specifically, the generation of CoT paths for spatial reasoning tasks is performed using LLMs, which allows us to capture rich intermediate reasoning steps that go beyond simple question-answer pairs.

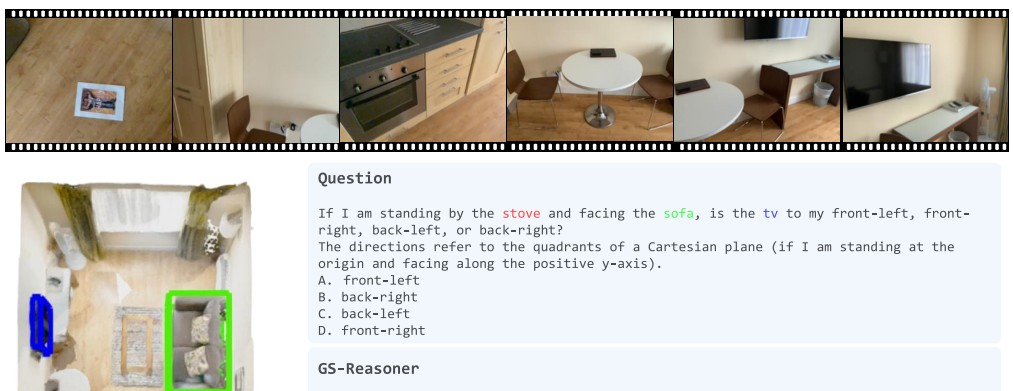

**Question**

If I am standing by the `stove` and facing the `sofa`, is the `tv` to my front-left, front-right, back-left, or back-right?
The directions refer to the quadrants of a Cartesian plane (if I am standing at the origin and facing along the positive y-axis).
A. front-left
B. back-right
C. back-left
D. front-right

**GS-Reasoner**

`<think>`The question asks for the TV's relative direction from my position (at the stove) while facing the sofa. I can find these objects as follows first:
stove 1 `<bbox>`(-1.8902, -1.7634, -1.0509, -1.3120, -0.9420, -0.1400)`</bbox>`, sofa 1 `<bbox>`(0.2240, 1.1322, -1.0505, 1.2367, 2.8400, -0.2070)`</bbox>`, tv 1 `<bbox>`(-1.9141, 1.7940, 0.0661, -1.8002, 2.5420, 0.6670)`</bbox>`

The relative direction can be determined by the local coordinates of the TV, calculated in 3 steps:
Step 1: Observer frame
Stove centroid (observer): (-1.601, -1.353), Sofa centroid: (0.730, 1.986), Facing vector: (2.332, 3.339), normalized → forward = (0.574, 0.822), Right vector = (0.822, -0.574)
Step 2: TV position
TV centroid: (-1.857, 2.168), Vector (observer → TV): (-0.256, 3.521)
Step 3: Projection to local coordinates
x_local = -2.269, y_local = 2.760. y_local > 0 → front; x_local < 0 → left. Among options A is correct.`</think>`
`<answer>`A`</answer>`

Figure 5: **Qualitative results on VSI-Bench.**

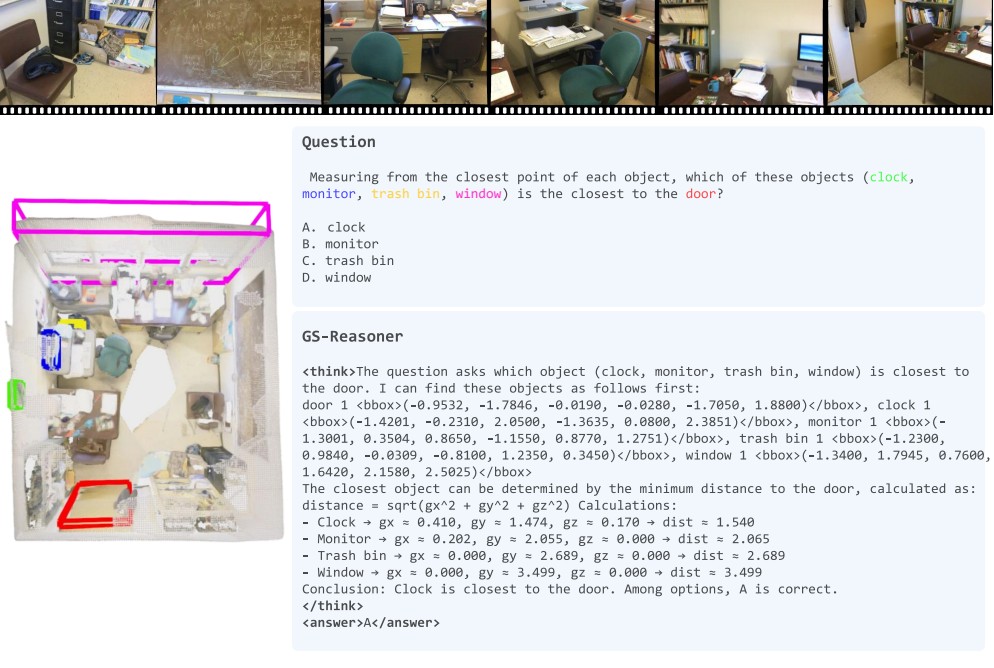

**Question**

Measuring from the closest point of each object, which of these objects (`clock`, `monitor`, `trash bin`, `window`) is the closest to the `door`?

A. clock
B. monitor
C. trash bin
D. window

**GS-Reasoner**

`<think>`The question asks which object (clock, monitor, trash bin, window) is closest to the door. I can find these objects as follows first:
door 1 `<bbox>`(-0.9532, -1.7846, -0.0190, -0.0280, -1.7050, 1.8800)`</bbox>`, clock 1 `<bbox>`(-1.4201, -0.2310, 2.0500, -1.3635, 0.0800, 2.3851)`</bbox>`, monitor 1 `<bbox>`(-1.3001, 0.3504, 0.8650, -1.1550, 0.8770, 1.2751)`</bbox>`, trash bin 1 `<bbox>`(-1.2300, 0.9840, -0.0309, -0.8100, 1.2350, 0.3450)`</bbox>`, window 1 `<bbox>`(-1.3400, 1.7945, 0.7600, 1.6420, 2.1580, 2.5025)`</bbox>`
The closest object can be determined by the minimum distance to the door, calculated as:
distance = sqrt(gx^2 + gy^2 + gz^2) Calculations:
- Clock → gx ≈ 0.410, gy ≈ 1.474, gz ≈ 0.170 → dist ≈ 1.540
- Monitor → gx ≈ 0.202, gy ≈ 2.055, gz ≈ 0.000 → dist ≈ 2.065
- Trash bin → gx ≈ 0.000, gy ≈ 2.689, gz ≈ 0.000 → dist ≈ 2.689
- Window → gx ≈ 0.000, gy ≈ 3.499, gz ≈ 0.000 → dist ≈ 3.499
Conclusion: Clock is closest to the door. Among options, A is correct.
`</think>`
`<answer>`A`</answer>`

Figure 6: **Qualitative results on VSI-Bench.**

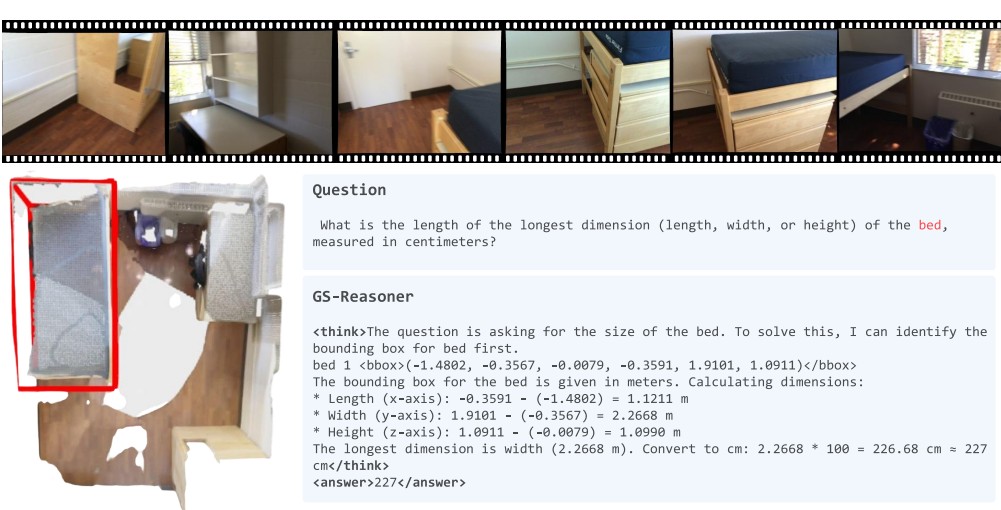

Figure 7: **Qualitative results on VSI-Bench.**

