# OpenReview forum: "Reasoning in Space via Grounding in the World"
_ICLR.cc/2026/Conference — ICLR 2026 Poster_

### Official Review · Reviewer_TRwm · 2025-10-31

**Soundness:** 3
**Presentation:** 3
**Contribution:** 3
**Rating:** 6
**Confidence:** 4

**Summary:**

This paper introduces GS-Reasoner, a novel spatial reasoning framework that enhances vision-language models' spatial understanding by incorporating 3D visual grounding as an intermediate reasoning step. Its core contributions include: a semantic-geometric hybrid 3D scene representation that aligns and fuses semantic features, geometric features, and 3D positional information via a dual-path pooling mechanism; the GCoT dataset, which provides 3D bounding box annotations and chain-of-thought reasoning paths to integrate grounding into spatial reasoning; and an autoregressive grounding capability that, enables end-to-end 3D localization without external detectors or modules, achieving competitive performance across multiple benchmarks.

**Strengths:**

1.The paper is well written and easy to follow
2.The dual-path pooling mechanism effectively mitigates misalignment issues between semantic-geometric and position-geometric features.
3.The GCoT dataset fills a critical gap in existing resources for integrated "grounding-reasoning" tasks, and extensive evaluations across 3D grounding, spatial reasoning, and general 3D tasks provide comprehensive validation.

**Weaknesses:**

1.The training stage is mainly focus on 3d reasoning,  I'm curious about whether the performance of GS-Reasoner on general-purpose benchmarks would decline after being trained on such a large amount of 3D-related data, and if so, by how much.
2.The construction pipeline of this representation method is somehow complex. It requires parallel execution of a 2D vision encoder, a 3D point cloud encoder, and an additional dual-path pooling fusion module, I wonder what is the. compute cost of these modules.
3.The "geometric" component of the entire representation heavily relies on the quality of depth maps and point clouds generated by preceding steps. In real-world scenarios, depth estimation (e.g., via VGGT-SLAM or MoGe-2) is inherently imperfect, I think it will be better to show how these noise will affect the performance of model.

**Questions:**

See Weakness.

---

> ### Author Response · Authors · 2025-11-22
> **Response to Reviewer TRwm (1/2)**
>
> We thank reviewer TRwm for positive feedback on our paper written, proposed strategy, GCoT dataset and comprehensive experiments. Our detailed responses to the comments are provided below.
>
> **[W1] GS-Reasoner performance on general-purpose benchmarks**
>
> Thank you for raising this concern. The backbone of GS-Reasoner (LLaVA-Next-Video) is a video-understanding model. Accordingly, we selected the ActivityNet-QA [1] and NExT-QA-MC [2] benchmarks, both commonly used in video understanding, to evaluate the general-purpose reasoning performance of GS-Reasoner. Since these benchmarks do not provide depth or pose annotations and include many dynamic objects, we use the TTT3R [3] model to estimate depth and pose for point-cloud construction, which shows better robustness in dynamic scenes compared with VGGT-SLAM. The results of GS-Reasoner on the two benchmarks are presented below:
>
> |  | NExT-QA-MC | AcNet-QA |
> | --- | --- | --- |
> | LLaVA-NeXT-Video | 80.4 | 52.5 |
> | GS-Reasoner | 70.0 | 34.0 |
>
> As observed, the performance of GS-Reasoner shows a certain decline compared with the backbone model, which we consider to be reasonable. Incorporating a new modality into the VLM input naturally affects its performance on the original tasks. In addition, our model is fine-tuned on a large amount of 3D reasoning data, which may further shift its behavior. Moreover, reconstructing point clouds from dynamic videos is highly challenging. Although TTT3R [3] performs better than other available approaches, its predicted depth and camera poses still contain substantial noise, which can influence the performance of GS-Reasoner on general tasks.
>
> The simplest way to address this issue is to mix in some of the pretrained VLM's SFT-stage data during model fine-tuning, similar to what some unified 2D and 3D understanding models do (e.g., LLaVA-3D [4], SR-3D [5]). Training with mixed data allows the model to focus more on image information when handling general video understanding tasks, thereby mitigating the negative impact caused by noise in 3D information. At the same time, it can still leverage 3D information to enhance spatial understanding when addressing spatial reasoning tasks. Therefore, we attempt to mix in a portion of LLaVA-Next-Video's SFT-stage data for GS-Reasoner training. Due to time constraints, we only use the NeXTQA training data here. We first preprocess all NeXTQA training videos using TTT3R to obtain depth and pose, resulting in a total of 3K videos and 17K QA pairs. We then mix this data with GCoT for fine-tuning, and the results are as follows:
>
> |  | NExT-QA-MC | VSIBench |
> | --- | --- | --- |
> | LLaVA-NeXT-Video | 80.4 | 35.6 |
> | GS-Reasoner (finetuned on GCoT) | 70.0 | 64.7 |
> | GS-Reasoner (finetuned on GCoT + 2D NeXTQA) | 79.1 | 65.2 |
>
> As shown, after training with the mixed data, GS-Reasoner significantly improves its general video understanding capabilities while maintaining nearly the same level of performance on spatial reasoning tasks. Our architecture supports the incorporation of more 2D data to further enhance the model's general video understanding capabilities, which would only require some engineering efforts.
>
>
>
>
> **[W2] Computation cost of GS-Reasoner**
>
> Please refer to the "General Response for Common Concerns" for a detailed answer.

---

> ### Author Response · Authors · 2025-11-22
> **Response to Reviewer TRwm (2/2)**
>
> **[W3] How noise in predicted depth maps and poses affects the performance of GS-Reasoner**
>
> We thank the reviewer for raising this insightful question. In fact, during our research phase, we experimented with various models for predicting depth and pose, ultimately selecting the VGGT-SLAM+MoGe2 approach. Through our experiments, we observed that two key factors significantly impact the performance of GS-Reasoner. Below are the test results on VSIbench using depth and pose predicted by different models(including latest proposed TTT3R[3] models):
>
> |  | AVG | Obj. Cnt. | Abs. Dist. | Obj. Size | Room Size | Rel.Dist. | Rel.Dir. | Route Plan | Appr. Order
> | --- | --- | --- | --- | --- | --- | --- | --- | --- | --- |
> | gt depth | 70.1 | 70.9 | 73.6 | 77.8 | 81.8 | 70.6 | 90.5 | 42.8 | 52.6
> | vggt-slam pred, moge2 metric align | 64.7| 69.1 | 61.9 | 70.0 | 65.7 | 65.4 | 88.9 | 44.3 | 52.3
> | vggt-slam pred, gt depth metric align | 67.8 | 69.4 | 68.9 | 76.5 | 78.8 | 65.2 | 87.6 | 43.4 | 52.9
> | vggt pred, moge2 metric align | 59.6 | 66.8 | 55.6 | 65.5 | 63.1 | 58.3 | 77.3 | 34.0 | 51.6
> | cur3r pred, no metric align | 56.8 | 66.8 | 55.3 | 60.2 | 50.7 | 59.8 | 76.6 | 34.0 | 50.7
> | ttt3r pred, no metric align | 58.0 | 67.5 | 57.6 | 60.2 | 46.5 | 62.8 | 82.9 | 35.1 | 51.5
>
> From the experiments, we summarize two key factors that influence performance:
> * **Accuracy of metric scale**. As observed, the most significant performance drop compared to using ground truth depth occurs in tasks related to absolute size estimation, such as room size and object absolute distance. During training, GS-Reasoner utilizes ground truth depth, which leads the model to prioritize 3D features in the input for these tasks. Consequently, inaccuracies in metric scale have a substantial impact on the results of such tasks. Notably, when using ground truth depth instead of MoGe2-predicted depth for metric alignment, there is a significant performance improvement.
> * **Accuracy of pose**. The accuracy of pose affects the relative positional distribution of object point clouds within the scene, thereby influencing the model's understanding of spatial relationships between objects. In the point clouds estimated by CUT3R [6], we observed significant pose errors, often resulting in completely incorrect scene layouts. This leads to a noticeable decline in GS-Reasoner's performance on tasks such as object relative direction. TTT3R improves upon CUT3R by reducing pose error accumulation during long-sequence inputs, resulting in some performance gains.
>
> Beyond these two factors, other aspects have a relatively minor impact on spatial reasoning. For instance, the quality of depth details (e.g., sharpness) does not significantly affect GS-Reasoner.
>
>
> [1] Yu, Zhou, et al. "Activitynet-qa: A dataset for understanding complex web videos via question answering." Proceedings of the AAAI Conference on Artificial Intelligence. Vol. 33. No. 01. 2019.
>
> [2] Xiao, Junbin, et al. "Next-qa: Next phase of question-answering to explaining temporal actions." Proceedings of the IEEE/CVF conference on computer vision and pattern recognition. 2021.
>
> [3] Chen, Xingyu, et al. "Ttt3r: 3d reconstruction as test-time training." arXiv preprint arXiv:2509.26645 (2025).
>
> [4] Zhu, Chenming, et al. "Llava-3d: A simple yet effective pathway to empowering lmms with 3d capabilities." Proceedings of the IEEE/CVF International Conference on Computer Vision. 2025.
>
> [5] Cheng, An-Chieh, et al. "3d aware region prompted vision language model." arXiv preprint arXiv:2509.13317 (2025).
>
> [6] Wang, Qianqian, et al. "Continuous 3d perception model with persistent state." Proceedings of the Computer Vision and Pattern Recognition Conference. 2025.

---

> > ### Comment · Reviewer_TRwm · 2025-11-27
> >
> > Thank you for your reply! I have no other questions and I will keep my positive rate of the paper.

---

> > > ### Author Response · Authors · 2025-11-27
> > > **Response to Reviewer TRwm**
> > >
> > > Thank you for your careful reading of our manuscript and rebuttal, as well as for your thoughtful and positive feedback. Your insights are invaluable and have significantly contributed to improving the quality of our work.
> > >
> > > Once again, we sincerely appreciate your kind guidance.

---

### Official Review · Reviewer_L2UB · 2025-11-01

**Soundness:** 2
**Presentation:** 3
**Contribution:** 2
**Rating:** 4
**Confidence:** 3

**Summary:**

This paper proposes GS-Reasoner, and a GCoT dataset, to tackle 3D visual grounding. Specifically, GS-Reasoner encodes image and 3D representations with a semantic geometric fusion model before feeding into a Video LLM for finetuning. GCoT dataset, on the other hand, construct QA pairs, and then augment them with CoT paths with GPT-4o based on the information of the bird’s eye view, object information, and Q&A. GS-Reasoner, when fine-tuned with GCoT, performed better on multiple visual grounding datasets, including ScanRefer, Multi3DRef, and spatial reasoning benchmarks like VSI-Bench, etc.

**Strengths:**

I think the architecture itself is a great contribution. Fusing depth and image seems like a straightforward but valid approach to improve spatial reasoning. I do wonder whether other information like normal maps could potentially give similar results.

For Table 2, I think showing the results of predicted and GT depth is a great addition, as it allows us to understand the upper bound of the current architecture and training.

Baselines are strong, spanning across closed and open-source models, as well as expert VLMs, with recent state-of-the arts such as VLM-3R-7B.

**Weaknesses:**

One of the main claims of this paper is that 3D visual grounding is the cornerstone of spatial reasoning. This suggests that improvement on spatial grounding would generally improve Spatial VQA, and from this perspective I find just the set of benchmarks tested is slightly lacking given the large variety of spatial reasoning benchmarks these days. To my understanding, SQA3D, Scan2Cap, and ScanQA questions are still majorly descriptors/grounding of 3D objects in a scene in similar scale. VSI-Bench is also heavily based on ScanNet and ScanNet++. The full story of spatial reasoning may not be told with these datasets alone. Some additional benchmarks that could be helpful (not asking for all evaluation but some additional ones with larger domain shift): SPAR-Bench, All-Angles Bench, MMSI-Bench, etc. This would give us insights on better visual grounding that translates to general 3D understanding.

The same questions also apply to training with/without GCoT and with/without CoT within the GCoT dataset. I believe understanding these would make the paper more comprehensive and solidify the claim that visual grounding is highly correlated with the other spatial reasoning tasks with larger domain change.

**Questions:**

The main questions I have derived from the Weaknesses.

1. Is GS-Reasoner set up purely for visual grounding? Or does it actually help with more generalized spatial reasoning to other scales and other types of questions?

2. Does fine tuning on the GCoT dataset help with visual grounding? Does it help with other spatial reasoning benchmarks?

3. Does CoT within the GCoT dataset help with visual grounding and other benchmarks?

Overall, I think the current stage of the experiments are not comprehensive for me to accept just yet. I hope the authors can shed light on some of the questions listed above.

---

> ### Author Response · Authors · 2025-11-22
> **Response to Reviewer L2UB (1/3)**
>
> We thank reviewer L2UB for recognizing the effectiveness of our proposed architecture and the clarity of our experimental results. Our detailed responses to the comments are provided below.
>
> For clarity, we organize our rebuttal into two parts, each addressing one of the key concerns: **R1. The claim that visual grounding is the cornerstone of spatial reasoning** and **R2. Responses to questions concerning the influence of GCoT on visual grounding**.
>
>
> ## **R1. The claim that visual grounding is the cornerstone of spatial reasoning (1/2)**
>
> **[W] Concerns regarding our claim that visual grounding is the cornerstone of spatial reasoning, and the need for additional evaluation on spatial reasoning tasks with larger domain shifts.**
>
> Thanks for raising this concern. We would first like to clarify our view on the relationship between visual grounding and spatial reasoning:
> * A VLM’s ability to perform visual grounding without relying on external modules reflects its inherent understanding of three-dimensional spatial structures;
> * Building on this, enhancing a model's visual grounding capability represents an improvement in its comprehension of three-dimensional space. In this context, for relatively simple spatial reasoning tasks (e.g., estimating room size), substantial performance gains can be achieved even without grounded chain-of-thought. However, for more complex tasks (e.g., inferring relative object directions), simply strengthening visual grounding does not necessarily yield proportional improvements. **The crucial factor is enabling the model to learn how to leverage visual grounding to support higher-level spatial reasoning.**
>
> Returning to our claim, we believe the assertion "spatial grounding would generally improve spatial VQA performance" remains valid. However, to fully prove this claim and to observe performance gains across a broad range of spatial reasoning tasks, the fairest experiments should be:
>
> * **Exp1**: LLaVA-Next-Video (backbone) trained on GCoT++ (w/o CoT)
> * **Exp2**: GS-Reasoner trained on GCoT++ (w/o CoT)
> * **Exp3**: GS-Reasoner trained on GCoT++ (w CoT)
>
> Here, the GCoT++ dataset should **encompass a broader range of reasoning tasks than GCoT, enabling the model to selectively leverage visual grounding to assist in reasoning across various spatial reasoning challenges**, and constructing such a dataset requires collective efforts from the research community. The expected outcomes of the above experiments are:
> * Exp2 $>$ Exp1, indicating that the GS-Reasoner’s superior 3D understanding can enhance relatively simple spatial reasoning performance without CoT;
> * Exp3 $>$ Exp2, demonstrating that with improved visual grounding capabilities, leveraging grounded CoT can further enhance relatively complex spatial reasoning performance.
>
> In our main paper, we constructed a relatively small dataset (GCoT) and tested it only on VSIBench to conduct these three experiments and validate our claim.
>
> So how does the GS-Reasoner trained on GCoT perform on other spatial reasoning benchmarks? To be honest, its performance is relatively poor. We tested the performance of GS-Reasoner on SPAR-bench, and the results are as follows (Note that low, medium, and high refer to the spatial reasoning levels defined in SPAR-Bench, each corresponding to a specific set of tasks):
>
> | | Avg | Low | Medium | High
> | --- | --- | --- | --- | ---
> | GS-Reasoner (ft GCoT w/o CoT) | 43.3 | 38.5 | 29.6 | 52.9
> | GS-Reasoner (ft GCoT w CoT) | 21.1 | 3.9 | 28.6 | 37.1
>
> As shown, training with CoT from GCoT actually results in worse performance compared to not using CoT. **However, this does not contradict our claim. The relatively weak results are primarily attributable to the limitations of the GCoT dataset.** Although the dataset is of high quality, its diversity and scale are still insufficient—covering only eight types of spatial reasoning tasks, similar to VSIbench—which is inadequate for teaching the model how to leverage visual grounding to support reasoning across a wide range of spatial reasoning challenges. Therefore, this experiment does not provide a definitive conclusion on whether visual grounding aids in more diverse spatial reasoning tasks.

---

> ### Author Response · Authors · 2025-11-22
> **Response to Reviewer L2UB (2/3)**
>
> ## **R1. The claim that visual grounding is the cornerstone of spatial reasoning (2/2)**
>
> To further validate our claim, we expand the GCoT dataset to create GCoT+. Specifically, we incorporate a subset (approximately 3M samples) of the SPAR-RGBD-7M dataset from SPAR-Bench into our GCoT. Note that we exclude the position matching task from the SPAR data, as our backbone model, LLaVA-Next-Video, can only accept fixed-resolution images, and the preprocessing of images would affect the outputs for this task. Due to time constraints, we are unable to construct detailed Grounded CoTs for the full SPAR training set. Therefore, we select two tasks—spatial\_imagination\_oc and spatial\_imagination\_oo tasks, both of which exhibit a large domain gap relative to VSIbench—and generate 3D bounding boxes annotations by projecting the provided 2D bounding boxes into 3D space and matching them with the ground-truth 3D bounding boxes in each scene. **We directly use these 3D bounding boxes along with their corresponding object names as a simplified form of Grounded CoT (without textual reasoning based on the bounding boxes)**, training the model to output the 3D bounding boxes before providing the final answer to assist in spatial reasoning. We retrained the three experiments mentioned above on the GCoT+ dataset, and the results on SPAR-bench are as follows (we use gt depth in evaluation):
>
> | | Avg | Depth-OC | Depth-OC-MV | Depth-OO | Depth-OO-MV | Dist-OC | Dist-OC-MV | Dist-OO | Dist-OO-MV | CamMotion | ViewChgI | DistI-OO | DistI-OO-MV | ObjRel-OC-MV | ObjRel-OO | ObjRel-OO-MV | SpImag-OC | SpImag-OC-MV | SpImag-OO | SpImag-OO-MV
> | --- | --- | --- | --- | --- | --- | --- | --- | --- | --- | --- | --- | --- | --- | --- | --- | --- | --- | --- | --- |  ---
> | LLaVA-Next-Video (GCoT+ w/o CoT) | 76.3 | 83.3 | 80.2 | 42.3 | 41.4 | 79.9 | 79.4 | 68.8 | 65.0 | 86.3 | 51.2 | 83.6 | 80.7 | 91.5 | 89.2 | 89.1 | 82.1 | 80.8 | 84.8 | 89.3
> | GS-Reasoner (GCoT+ w/o CoT) | 81.9 | 88.1 | 84.2 | 50.3 | 45.8 | 86.6 | 85.6 | 73.5 | 72.6 | 92.3 | 72.7 | 87.4 | 87.5 | 94.0 | 90.9 | 93.6 | 86.8 | 87.2 | 84.4 | 93.3
> | GS-Reasoner (GCoT+ w CoT) | - | - | - | - | - | - | - | - | - | - | - | - | - | - | - | - | 88.4 | 90.1 | 89.7 | 95.2
>
>
> As shown, after training on the expanded GCoT+ dataset and evaluating on the broader spatial reasoning tasks in SPAR-Bench, we continue to observe the expected trends: Exp2 outperforms Exp1, and Exp3 further improves upon Exp2. Moreover, in Exp3, **even the simplified Grounded CoT leads to measurable improvements** on the spatial\_imagination\_oc and spatial\_imagination\_oo tasks, particularly on the spatial\_imagination\_oo task, further validating our claim.
>
> Back to the questions:
>
> **[Q1] Is GS-Reasoner set up purely for visual grounding? Or does it actually help with more generalized spatial reasoning to other scales and other types of questions?**
>
> The GS-Reasoner is not set up purely for visual grounding; rather, it is designed to genuinely enhance the 3D understanding capabilities of VLMs. The ability to perform visual grounding without relying on external modules serves as strong evidence of the model's 3D comprehension. Furthermore, as shown in Tab above, the results of Exp2 outperforming Exp1 indicate that even **without** Grounded CoT, the GS-Reasoner achieves better spatial reasoning results compared to the backbone model, thanks to its superior spatial understanding capabilities.
>
>
> **[Q2 \& Q3] GCoT influence across other spatial reasoning benchmarks**
>
> Regarding the influence of GCoT on other spatial reasoning benchmarks, we have provided a detailed explanation above. In summary, the scale and diversity of the GCoT dataset are still insufficient to teach the model to correctly utilize grounding for reasoning across various spatial reasoning tasks. Therefore, regardless of whether the model is trained on GCoT or whether CoT is used, the results on SPAR-Bench cannot definitively confirm the validity of our claim. Consequently, we further constructed the GCoT+ dataset and retrained the models on this dataset, with results confirming our expected trends.
>
> Finally, we agree with the reviewer that “the full story of spatial reasoning may not be told with these datasets alone.” Spatial reasoning covers a wide range of abilities. For tasks involving spatial relations (e.g., object relationships, directions, distances) and spatial imagination (e.g., room size, object size, scene layout)—which we consider especially relevant for embodied intelligence—visual grounding is highly important. In contrast, for more abstract spatial reasoning (such as space folding or geometric reasoning in OmniSpatial [1]), visual grounding may play a more limited role. Thus, a more accurate statement is that “visual grounding is **one of the cornerstones** of spatial reasoning for VLMs.” We again thank the reviewer for the helpful feedback, which enabled us to clarify our position.

---

> ### Author Response · Authors · 2025-11-22
> **Response to Reviewer L2UB (3/3)**
>
> ## **R2. GCoT influence on visual grounding**
> **[Q2 \& Q3] GCoT influence on visual grounding**
>
> Regarding this question, we would first like to clarify that the chain-of-thought (CoT) component in GCoT is specifically designed to serve spatial reasoning tasks and is not intended to enhance visual grounding capabilities. This CoT component begins by analyzing the question to determine whether visual grounding is necessary to assist in reasoning. If grounding is required, it identifies all objects mentioned in the question and grounds them in 3D space, subsequently using this grounding information to perform spatial reasoning before providing the final answer. The visual grounding task involved in this process is essentially phrase grounding, which entails locating the corresponding object positions in 3D space based on the phrases referring to objects in the question. This task is relatively easier compared to the sentence grounding task in ScanRefer, as it does not require considering context in the sentence.
>
>
> Based on the above clarification, we can better explain the following experimental results:
>
> |  | ScanRefer Acc@25 | ScanRefer Acc@50 | Multi3DRef F1@25 | Multi3DRef F1@50
> | --- | --- | --- | --- | ---
> | GS-Reasoner (not finetuned on GCoT) | 60.0 | 41.3 | 60.6 | 42.7
> | GS-Reasoner (finetuned on GCoT, w/o CoT) | 60.3 | 41.7 | 61.4 | 43.5
> | GS-Reasoner (finetuned on GCoT, w CoT) | 60.8 | 42.2 | 61.7 | 45.3
>
> First, when fine-tuning the model on GCoT without CoT, we observe some improvements in visual grounding tasks, indicating that enhancements in the model's spatial reasoning capabilities can aid visual grounding to some extent. However, the improvements are modest. We believe this is primarily due to the presence of many ambiguous descriptions in ScanRefer and Multi3DRefer, such as not specifying the viewpoint when describing left and right. This issue is also highlighted in BEACON3D [2]. Consequently, even with improved spatial understanding, the model cannot fully resolve these ambiguities, leading to limited performance gains.
>
> Second, when fine-tuning the model on GCoT with CoT, we observe further improvements. As mentioned earlier, the CoT component in GCoT is designed for spatial reasoning tasks and is not utilized during the visual grounding task. Therefore, the observed improvements are not due to enhanced spatial understanding. We believe that the inclusion of ScanNet++ data in GCoT contributes to this improvement. Compared to the ScanNet dataset used in ScanRefer and Multi3DRefer, ScanNet++ offers more diverse and richly annotated 3D scenes and objects. Training with CoT on this more varied grounding task enhances the model's visual grounding capabilities.
>
>
>
> [1] Jia, Mengdi, et al. "OmniSpatial: Towards Comprehensive Spatial Reasoning Benchmark for Vision Language Models." arXiv preprint arXiv:2506.03135 (2025).
>
> [2] Huang, Jiangyong, et al. "Unveiling the mist over 3d vision-language understanding: Object-centric evaluation with chain-of-analysis." Proceedings of the Computer Vision and Pattern Recognition Conference. 2025.

---

### Official Review · Reviewer_2vrB · 2025-11-01

**Soundness:** 3
**Presentation:** 3
**Contribution:** 3
**Rating:** 6
**Confidence:** 4

**Summary:**

This paper introduces a unified representation of geometry and semantics, and proposes a method called Dual-Path Pooling to address misalignment issues in deriving per-patch representations. Furthermore, the authors present the GCoT dataset, which incorporates grounding as an intermediate step in spatial reasoning to enhance the spatial reasoning capabilities of MLLMs.

**Strengths:**

1. The two identified misalignments are a valid concern, and the proposed Dual-Path Pooling offers a simple yet effective solution to mitigate them. In particular, the approach of directly sampling 3D points and subsequently interpolating their geometric features is both elegant and effective.

2. I agree that grounding is essential for spatial reasoning, and the proposed GCoT appears to be of high quality.

3. The experiments are sufficient to demonstrate the effectiveness of the proposed method and dataset.

**Weaknesses:**

1. The paper claims that the lack of a unified 3D representation leads to reliance on external modules. However, as shown in Table 1, while the proposed method significantly improves the Acc@25 metric on ScanRefer, it does not outperform LLMs equipped with external grounding modules on Acc@50. This suggests that although the method enhances the model’s spatial reasoning capabilities, it does not improve its ability to perform precise object localization—and thus does not fully eliminate the need for additional localization modules. Accurate localization may still require such external modules.

2. The efficiency of the proposed model is unclear; it would be beneficial to include an analysis or empirical results demonstrating its computational efficiency.

3. Since GCoT is built upon GPT-4o, it is important to clarify how the authors ensure the correctness of the generated data. Providing a explanation of the validation or filtering mechanisms would help substantiate the quality of the proposed dataset and, in turn, strengthen its overall contribution.

**Questions:**

The issues identified and the proposed solutions appear reasonable, and the introduced dataset makes a meaningful contribution. While a few minor points remain unclear, I believe they can be adequately addressed during the rebuttal phase.

---

> ### Author Response · Authors · 2025-11-21
> **Response to Reviewer 2vrB**
>
> We thank the reviewer 2vrB for acknowledging effectiveness of proposed strategy, our main claim and the clarity of our experimental results. Our detailed responses to the comments are provided below.
>
> **[W1] Precise object localization ability in visual grounding**
>
> Please refer to the "General Response for Common Concerns" for a detailed answer.
>
> **[W2] Computation efficiency of GS-Reasoner**
>
> Please refer to the "General Response for Common Concerns" for a detailed answer.
>
> **[W3] GCoT data filtering strategy**
>
> Thanks for raising this concern. In fact, we do not employ a complex filtering strategy during the data construction process. We only use GPT-4o to generate Chain-of-Thought paths, while the original QA without CoT is generated based on templates, which ensures the correctness of the QA part. When prompting GPT to generate CoT, in addition to the BEV, Question, and Answer mentioned in the paper, we also provide one CoT example as context for each type of question to ensure that GPT generates reasonable CoTs as much as possible.
>
> During the initial phase of constructing a small amount of data for validation, we did find that GPT sometimes failed to generate reasonable CoTs, i.e., the CoT could not lead to the correct answer. GPT itself would also recognize this and indicate in its output that the answer could not be derived from the CoT. Therefore, when generating data on a large scale later, we added a prompt instructing GPT to directly output "unable to generate a reasonable CoT" when it finds that it cannot generate a reasonable CoT. We then filter out these samples through simple string matching and resend them to GPT for generation. In most cases, GPT is able to generate reasonable CoT paths, so the number of repeated generations is only 1-2 times.

---

> > ### Comment · Reviewer_2vrB · 2025-11-28
> > **Final review**
> >
> > Thanks for the response, which has almost addressed my concerns.  I have no other questions. I will maintain my positive rating.

---

### Official Review · Reviewer_ckhy · 2025-11-07

**Soundness:** 3
**Presentation:** 4
**Contribution:** 3
**Rating:** 8
**Confidence:** 3

**Summary:**

This paper proposes a 3D large language model, GS-Reasoner, and introduces a new dataset, GCoT. The GS-Reasoner constructs a semantic–geometric hybrid representation of 3D scenes through a dual-path pooling mechanism. The model achieves strong performance on both 3D visual grounding and spatial reasoning tasks without relying on pretrained 3D detectors or external decoders.

**Strengths:**

1. Clear motivation addressing key limitations of 3D LLMs. The work is both meaningful and timely, as it explores how to empower 3D LLMs with spatial reasoning and visual grounding capabilities without depending on pretrained 3D detectors or external decoders.
2. Well-presented methodology and thorough validation. The paper provides detailed methodological descriptions and extensive experiments. The results are competitive on 3D visual grounding tasks and achieve state-of-the-art performance on spatial reasoning and general 3D benchmarks.

**Weaknesses:**

Weaknesses:
1. Performance gap with state-of-the-art baselines. The proposed GS-Reasoner still lags behind ROSS3D on several key benchmarks, including ScanRefer (Acc@50), Multi3DRef (F1@50), ScanQA, and SQA3D.
2. Dependence on external modules for geometry estimation. The proposed GS-Reasoner still relies on VGGT-SLAM to estimate depth maps and camera parameters, which introduces additional dependencies and may limit the model’s end-to-end autonomy.

**Questions:**

1. The proposed GS-Reasoner shows a relatively larger performance gap between Acc@25 and Acc@50 on ScanRefer, and between F1@25 and F1@50 on Multi3DRef, compared with other methods. Could the authors clarify the reason behind this discrepancy? Additionally, what strategies might help narrow this gap?

---

> ### Author Response · Authors · 2025-11-21
> **Response to Reviewer ckhy**
>
> We thank reviewer ckhy for the positive feedback on our motivation, presentation, and comprehensive experiments. Our detailed responses to the comments are provided below.
>
> **[W1] Performance gap between GS-Reasoner and SOTA methods on ScanRefer(Acc@50), Multi3DRef(Acc@50), ScanQA and SQA3D**
>
> Thanks for raising this concern.
>
> Regarding the performance gap on visual grounding tasks (the precise reasoning gap on ScanRefer and Multi3DRef), we address the analysis in our response to Q1. Here, we focus on explaining the reasons behind the performance gap between GS-Reasoner and SOTA methods on ScanQA and SQA3D. We believe the main reasons are the presence of many ambiguous questions in these datasets that do not clearly specify the target object, as well as a strong bias in answer distribution. These factors encourage the model to overfit to textual patterns instead of effectively exploiting 3D tokens. Recent studies BEACON3D[1] and 3DRDQA[2] have also shown that finetuning LLMs without visual input can yield results comparable to the state of the art on SQA3D.
>
> We further test the effect of removing the geometry features and position encoding from GS-Reasoner, and then fine-tune the model on ScanQA and SQA3D. The results are as follows:
>
> | | ScanQA | SQA3D
> | --- | --- | ---
> |  | C↑, EM↑ | EM↑
> GS-Reasoner (w/o geo, w/o 3d pe) |  102.1, 30.0 | 59.8
> GS-Reasoner (full) |  102.6, 30.0 | 59.9
>
> We find that regardless of whether 3D information is included, the model's performance on ScanQA and SQA3D remains nearly unchanged. This indicates that the **model does not effectively utilize 3D information to enhance spatial understanding during training**, but rather relies more on textual patterns for answering.
>
> However, there are still some ways to improve the model's performance on ScanQA and SQA3D, such as introducing reconstruction constraints in the VLM's output, as done in ROSS3D [3], to encourage the model to utilize the 3D information in the input. How to integrate such constraints with our GS-Reasoner architecture is a promising direction for future research.
>
> **[Q1] Relatively larger performance gap on Acc@25 and Acc@50 in visual grounding tasks**
>
> Please refer to the "General Response for Common Concerns" section for a detailed answer.
>
>
>
>
> [1] Huang, Jiangyong, et al. "Unveiling the mist over 3d vision-language understanding: Object-centric evaluation with chain-of-analysis." Proceedings of the Computer Vision and Pattern Recognition Conference. 2025.
>
> [2] Li, Haoyuan, et al. "Does Your 3D Encoder Really Work? When Pretrain-SFT from 2D VLMs Meets 3D VLMs." arXiv preprint arXiv:2506.05318 (2025).
>
> [3] Wang, Haochen, et al. "Ross3d: Reconstructive visual instruction tuning with 3d-awareness." arXiv preprint arXiv:2504.01901 (2025).

---

### Author Response · Authors · 2025-11-21
**General Response for Common Concerns**

### **Relatively larger performance gap on Acc@25(F1@25) and Acc@50(F1@50) in visual grounding tasks**

We thank reviewers ckhy (Q1) and 2vrB (W1) for raising this question, and we provide a unified response here.

In fact, we have also noticed this issue during our research and have been trying to narrow this gap. One of the purposes of the dual-path pooling strategy proposed in the main paper is to reduce this performance gap. Dual-path pooling provides more accurate geometric features for image patches, enabling more precise localization. We have also validated the effectiveness of this strategy in our ablation studies(Tab. 5). It can be observed that the improvement in Acc@50 is significantly greater than that in Acc@25 when using this strategy.

However, as the reviewers mentioned, the gap between GS-Reasoner and baseline methods in Acc@25(F1@25) and Acc@50(F1@50) remains relatively larger, which also results in GS-Reasoner performing worse than some baselines in Acc@50(F1@50). We conduct a comparative analysis of the differences between GS-Reasoner and baseline methods and believe that the key issues lie in the following two aspects:

* **Mask supervision during visual grounding training**. Almost all baseline methods benefit from mask supervision, either directly (by using segmentation loss for training) or indirectly (by utilizing proposal priors, where proposals are extracted from mask3D trained with segmentation loss). Mask supervision is more conducive to precise object localization than bbox supervision, as also reported in Locate3D[1] and UniVLG[2].
* **Inference with mesh point cloud input**. Most baseline methods use mesh point cloud input during visual grounding inference, while GS-Reasoner only uses sensor point cloud. Compared to sensor point cloud, mesh point cloud has less noise and higher point cloud density, providing more accurate geometric information that helps improve localization accuracy.

In the context of GS-Reasoner, we cannot leverage mask supervision to enhance grounding accuracy. Therefore, a direct approach is to increase the number of point cloud features contained within the target object, thereby compensating for the inherent limitations of sensor point clouds. We have implemented two methods to achieve this:
* Increasing the number of input images, from 32 to 48 or 64;
* Using a frame sampling algorithm that is relevant to the question context. Here, we adopt the algorithm proposed in cdViews[3] for sampling.

By employing the above two methods, we achieved the following results on ScanRefer and Multi3DRef:

|  | ScanRefer Acc@25 | ScanRefer Acc@50 | Multi3DRef F1@25 | Multi3DRef F1@50 |
| --- | --- | --- | --- | --- |
| 32 frames uniform | 60.8 | 42.2 | 61.7 | 45.3 |
| 48 frames uniform | 61.7 | 44.5 | 62.2 | 46.9 |
| 64 frames uniform | 61.2 | 45.1 | 62.5 | 48.0 |
| 32 frames cdViews sample | 61.1  | 43.4 | 61.9 | 46.2 |

As can be seen, uniformly sampling more frames leads to a **more pronounced improvement in Acc@50 (F1@50) compared to Acc@25 (F1@25)**, indicating that providing denser point clouds can indeed help enhance localization accuracy. Similarly, using a frame sampling algorithm relevant to the question context also yields comparable effects.

We believe that achieving further improvements under this setting is inherently challenging. In particular, surpassing the sota baselines in Acc@50 (F1@50) without any post-processing is difficult, as bounding-box supervision is fundamentally less informative than mask supervision in 3D visual grounding tasks.

### **Computation efficiency of GS-Reasoner**
GS-Reasoner's computational overhead compared to the backbone LLaVA-Next-Video primarily arises from the construction of visual tokens. Specifically, GS-Reasoner requires encoding point clouds using Point Transformer and integrating them into image patch features. Therefore, we conducted a comparison of computational overhead during the visual token construction phase, using a single 4090 GPU with 32 input images. The results are as follows (in milliseconds):

|  | 2D vision encoder | 3D vision encoder | dual path pooling | total input preparation |
| --- | --- | --- | --- | --- |
| LLaVA-Next-Video | 429 | - | - | 470 |
| GS-Reasoner | 430 | 204 | 41 | 737 |


As can be seen, the additional overhead introduced in the 3D vision encoder and dual-path pooling is acceptable. Regarding the subsequent VLM autoregressive inference phase, since the number of constructed visual tokens remains unchanged, the computational overhead is the same as that of the backbone.


[1] McVay, Paul, et al. "LOCATE 3D: Real-World Object Localization via Self-Supervised Learning in 3D." ICML 2025.

[2] Jain, Ayush, et al. "Unifying 2D and 3D Vision-Language Understanding." ICML 2025.

[3] Wang, Fengyun, et al. "3D Question Answering via only 2D Vision-Language Models." ICML 2025.

---

### Author Response · Authors · 2025-11-28
**Official Comment by Authors**

Dear Reviewers,

We sincerely thank all reviewers for your insightful comments and recognition of our work.

As the discussion period is approaching its end with less than a week remaining, we wanted to ensure that we have addressed all of your concerns satisfactorily. If there are any additional comments or feedback you would like us to consider, please feel free to let us know. Your insights are invaluable to us, and we are eager to address any remaining issues to further improve our work.

Thank you very much for your time and effort in reviewing our paper.

---

### Author Response · Authors · 2025-12-02
**Summary Response to AC**

Dear Area Chair,

We would like to express our sincere gratitude for your considerable efforts throughout the review process. We also thank all reviewers for their constructive feedback and for the insightful discussions during the rebuttal period, which have helped us further improve the paper. Below, we provide a concise summary of the rebuttal discussion to assist in your final evaluation.

***1. Common positive feedback by reviewers***

Here, we summarize the common positive feedback highlighted by the reviewers:
* **Clear motivation.** "Clear motivation addressing key limitations of 3D LLMs"(`ckhy`), "I agree that grounding is essential for spatial reasoning"(`2vrB`).
* **Effective and well-designed architecture.** "The two identified misalignments are a valid concern, and the proposed Dual-Path Pooling offers a simple yet effective solution to mitigate them"(`2vrB`), "I think the architecture itself is a great contribution"(`L2UB`), "The dual-path pooling mechanism effectively mitigates misalignment issues"(`TRwm`).
* **Comprehensive experiments evaluation.** "thorough validation"(`ckhy`), "The experiments are sufficient"(`2vrB`), "extensive evaluations across 3D grounding, spatial reasoning, and general 3D tasks provide comprehensive validation"(`TRwm`)
* **Clear and well-structured writing.** "The paper provides detailed methodological descriptions"(`ckhy`), "The paper is well written and easy to follow"(`TRwm`)


***2. Concerns raised by reviewers***

Here, we summarize the concerns raised by the reviewers:
* Performance gap in ScanQA, SQA with SOTA methods (`ckhy`)
* Relatively larger performance gap on Acc@25(F1@25) and Acc@50(F1@50) in visual grounding tasks (`ckhy`, `2vrB`)
* Computation efficiency of GS-Reasoner (`2vrB`, `TRwm`)
* GCoT data filtering strategy (`2vrB`)
* More evaluation on spatial reasoning benchmarks to prove the claim that visual grounding is the cornerstone of spatial reasoning (`L2UB`)
* GCoT dataset influence on visual grounding (`L2UB`)
* Performance decline on general-purpose benchmarks (`TRwm`)
* The noise in depth and pose estimation influence on spatial reasoning (`TRwm`)

For the concerns raised, we have provided detailed experiments and analyses in our responses. Please refer to the corresponding “Response to Reviewer” sections below for specific discussions. **Following our clarifications, reviewers `2vrB` and `TRwm` indicated that their concerns have been fully addressed.**

Regarding reviewer `L2UB`’s comments in particular, we made substantial efforts to elaborate on our motivation, experimental setup, and the reasoning behind our claim. We also clarified why the additional experiments suggested by reviewer `L2UB` would not effectively test or support our claim. To further strengthen our position, we expanded the GCoT dataset and conducted larger-scale training and evaluation to validate the claim. Unfortunately, we did not have the opportunity to engage in further discussion with reviewer `L2UB` during the rebuttal period.

***3. Paper revision***

Based on the reviewers’ feedback, we have made the following revisions to the paper (highlighted in blue):
* Clarified our main claim more explicitly.
* Added more detailed analysis and discussion of the results on general 3D tasks.
* Added computational cost experiments in Tab.7 along with corresponding analysis.
* Added an ablation study on input frames in Tab.8 along with corresponding analysis.
* Added experiments on the impact of predicted depth and pose on spatial reasoning in Tab.9 along with corresponding analysis.

---
Once again, we sincerely appreciate the additional time and effort the Area Chair has devoted to evaluating our submission. We are also deeply grateful to all reviewers for their recognition of our work and for the constructive and insightful suggestions that have significantly helped improve our manuscript. We hope that this summary is helpful in supporting your assessment during the review process.

Best Regards,

Authors of Submission 1387.

---

### Meta-Review · Area_Chair_Xhoe · 2025-12-23

**Summary:**

This paper makes an argument that grounded 3D reasoning is required for spatial understanding.  The authors make two contributions.  First, they propose an approach for grounded 3D reasoning in an LLM without an external localization module.  Second, they introduce a grounded CoT benchmark to help evaluate their approach.  While the reviewers raised a few concerns, they still generally argued for acceptance.  None of the remaining concerns rise to the level of supporting an argument for rejection.

**Reviewer Concerns:**

A common question that arise was whether grounding was indeed required for spatial reasoning, with some larger gaps reported on some benchmarks.  The authors rebuttal largely validated this concern. The authors did make an argument about the challenges of improving performance and also the benefits that mask supervision may provide, but the AC finds these arguments shaky at best, with alternative explanations or not enough evidence to support strong conclusions.

**Reviewer Scores:**

I do not believe any of the discussion would likely have had a significant impact on reviewer scores.

---

### Decision · Program_Chairs · 2026-01-26

Accept (Poster)